# A Short Introduction to Cosmology and its Current Status

**Pedro G. Ferreira[1⋆] and Alexander Roskill[1†]**

**1** Astrophysics, University of Oxford, DW Building, Keble Road, Oxford OX1 3RH, UK

⋆ pedro.ferreira@physics.ox.ac.uk ,    † alexander.roskill@physics.ox.ac.uk

## Abstract

The current cosmological model, known as the $\Lambda$-Cold Dark Matter model (or $\Lambda$CDM for short) is one of the most astonishing accomplishments of contemporary theoretical physics. It is a well-defined mathematical model which depends on very few ingredients and parameters and is able to make a range of predictions and postdictions with astonishing accuracy. It is built out of well-known physics – general relativity, quantum mechanics and atomic physics, statistical mechanics and thermodynamics – and predicts the existence of new, unseen components. Again and again it has been shown to fit new data sets with remarkable precision. Despite these successes, we have yet to understand the unseen components of the Universe and there has been evidence for inconsistencies in the model. In these lectures, we lay the foundations of modern cosmology.

# 1 Preamble

The following set of lectures was designed for the Les Houches Summer School 2025 on The Dark Universe. The task was to summarise modern cosmology in four lectures of 1.5 hours each. This was quite a challenge as they have to cover a lot of material in too little time. They should be seen more as a roadmap which points the reader to the relevant topics. If anything, you might want to use these lectures to dive into more detailed explanations of the various topics that they touch on. Other lecturers at this Summer School will have gone into more depth in the various topics. We have relied on David Alonso's cosmology lecture notes and Pedro Ferreira's general relativity & cosmology lecture notes for Oxford students. You can find many other texts and lecture notes out there that you can consult [1–3].

We should get some conventions established from the outset. For a start, we will assume that the time coordinate, $t$, and the spatial coordinate, $\vec{x} = (x^1, x^2, x^3)$, can be organised into a 4-vector $(x^0, x^1, x^2, x^3) = (ct, \vec{x})$, where $c$ is the speed of light. In these lectures we will take $c = 1$, unless we find it useful to keep it in explicitly. Throughout these lecture notes we will use the $(-, +, +, +)$ convention for the metric. We will be using the convention that Roman labels (like $i$, $j$, etc) span 1 to 3 and label spatial vectors, while Greek labels (such as $\alpha$, $\beta$, etc) span 0 to 3 and label space-time vectors.

# 2 Introduction

Cosmology has undergone a dramatic transformation over the past few decades, evolving from a 'data-starved' science into what is now often called *precision cosmology* [4]. The field now describes in remarkable detail how the Universe has evolved from fractions of a second after the Big Bang to the present day, all encapsulated within the beautifully simple Λ-Cold Dark Matter (ΛCDM) model. As one of the greatest successes of theoretical and experimental physics, ΛCDM has consistently been shown to match an ever-growing body of observations, from increasingly distant galaxies to exquisitely precise measurements of the cosmic microwave background (Fig. 1).

One might say that ΛCDM is so accurate it is boring and there is nothing left to do, or to discover. That might be the case. Except that we have yet to fully understand the unseen com-

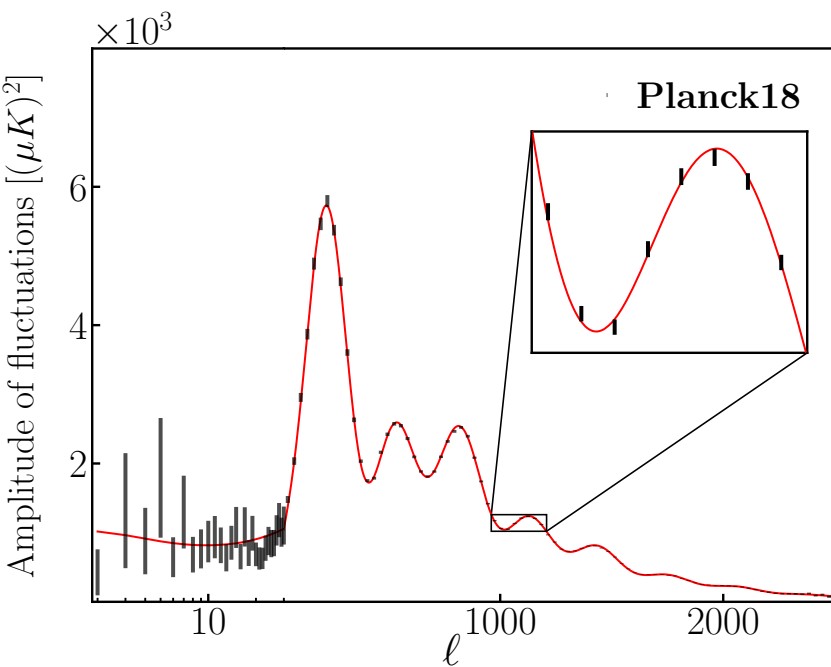

Figure 1: The Planck 2018 measurements [5] of the angular power spectrum of the cosmic microwave background (black lines) which show remarkable agreement with the theoretical predictions of the ΛCDM model (red line).

ponents (more of that later) and, furthermore, there is tentative evidence for inconsistencies in the model, i.e. different data sets are pushing the model in different directions. This means that cosmology is alive and kicking and needs to be worked on, or with, for a lot longer.

In these lectures we will lay down the foundations of, and describe in some detail, the mathematical model at the heart of modern cosmology. We hope to convince you that it is a remarkable achievement but also entice you to work on it and push it further.

## 3 A mathematical model for a homogeneous and isotropic universe

We have grown to believe that we don't live in a special place, that we are not at the centre of the Universe. Oddly enough, this point of view allows us to make some far-reaching assumptions. So, for example, if we are insignificant and, furthermore, everywhere is insignificant, then we can assume that at any given time, the Universe looks the same everywhere. In fact we can take that statement to an extreme and assume that at any given time, the Universe looks *exactly* the same at every single point in space. Such a space-time is dubbed to be *homogeneous*.

There is another assumption that takes into account the extreme regularity of the Universe and that is the fact that, at any given point in space, the Universe looks very much the same in whatever direction we look. Again, such an assumption can be taken to an extreme so that at any point, the Universe looks *exactly* the same, whatever direction one looks. Such a space-time is dubbed to be *isotropic*.

Homogeneity and isotropy are distinct yet inter-related concepts. For example a universe which is isotropic will be homogeneous, while a universe that is homogeneous *may not* be isotropic. A universe which is only isotropic around one point is not homogeneous. A universe that is both homogeneous and isotropic is said to satisfy the *Cosmological Principle*. It is

believed that our Universe satisfies the Cosmological Principle.

There are three homogeneous and isotropic space time metrics (flat, hyper-spherical, hyper-hyperbolic), which can be written in a unified way, in spherical coordinates:

$$ds^2 = -dt^2 + a^2(t)\left[\frac{dr^2}{1-kr^2} + r^2(d\theta^2 + \sin^2\theta d\phi^2)\right],$$

where $k$ is positive, zero or negative for spherical, flat or hyperbolic geometries, and $|k| = 1/R^2$, where $R$ is the scale of curvature of space. For a flat (Euclidean) Universe the metric looks particularly simple:

$$ds^2 = -dt^2 + a^2(t)[(dx^1)^2 + (dx^2)^2 + (dx^3)^2].$$

We call $a(t)$ the scale factor and $t$ is normally called *cosmic* time or *physical* time. By applying the Cosmological Principle, we have collapsed the 10 components of the space-time metric (which is a function of the 4 space-time coordinates), $g_{\alpha\beta}(x^\mu)$, into one function of time, $a(t)$.

This tells us about space-time. What about 'stuff'? To fully characterise 'stuff' we need a microphysical model, such as, for example, the standard model of particle physics or at least some atomistic model (to which we can apply statistical mechanics) or some field theory. But if we are looking at the large scale properties of space time, we can describe 'stuff' as a fluid with an energy density, $\rho$, a pressure $P$, and a 4-velocity, $U^\alpha$. These can be packed together to form the energy momentum tensor:

$$T^{\alpha\beta} = (\rho + P)U^\alpha U^\beta + Pg^{\alpha\beta},$$

where

$$U^\alpha U^\beta g_{\alpha\beta} = -1.$$

Assuming the Cosmological Principle we have

$$\begin{aligned} U^\alpha &= (1,0,0,0), \\ T_{00} &= \rho, \\ T_{ij} &= a^2 P \tilde{g}_{ij}, \end{aligned}$$

where $\tilde{g}_{ij}$ is the metric of space with the scale factor $a^2$ divided out.

We find the dynamics from the Einstein Field Equations

$$G^{\alpha\beta} = 8\pi G T^{\alpha\beta} - \Lambda g^{\alpha\beta},$$

where $G^{\alpha\beta} = R^{\alpha\beta} - \frac{1}{2}Rg^{\alpha\beta}$ is the Einstein tensor and $\Lambda$ is the cosmological constant. We can also use conservation of energy

$$\nabla_\alpha T^{\alpha\beta} = 0,$$

where $\nabla_\alpha$ is the covariant derivative associated with $g_{\alpha\beta}$. Note, however, that this equation is not independent from the Einstein field equations.

If we work through these equations [6–9], assuming the cosmological principle, we find that the scale factor obeys the Friedmann (or FRW for Friedmann-Robertson-Walker) equation

$$\left(\frac{\dot{a}}{a}\right)^2 = \frac{8\pi G}{3}\rho - \frac{k}{a^2} + \frac{\Lambda}{3},$$

and the Raychaudhuri equation

$$\frac{\ddot{a}}{a} = -\frac{4\pi G}{3}(\rho + 3P) + \frac{\Lambda}{3}.$$

We call $H_0 = \dot{a}/a(t_0)$ the *Hubble constant*, where $t_0$ is the cosmic time today.

If we want to solve them we need to have a model for $\rho(a)$ and $P(a)$ – there are too many unknowns. Short of having a microphysical model, and using statistical mechanics or field theory to find their dependence on $a$, we can embrace the idea that they are a fluid and define an equation of state, $w(a)$

$$w \equiv \frac{P}{\rho}.$$

The equation for conservation of energy then looks like

$$\dot{\rho} + 3\frac{\dot{a}}{a}[1 + w(a)]\rho = 0,$$

which can be integrated to give

$$\rho \propto e^{-3\int \frac{da}{a}[1+w(a)]}.$$

Let us now focus on a few specific examples, starting off with the case of *non-relativistic* matter. A notable example is that of massive particles whose energy is dominated by the rest energy of each individual particle. This kind of matter is sometimes simply called *matter* or *dust*. We can guess what the evolution of the mass density should be. The energy in a volume $V$ is given by $E = M$ so $\rho = E/V$, where $\rho$ is the mass density. But in an evolving Universe we have $V \propto a^3$, so $\rho \propto 1/a^3$. Alternatively, note that $P \simeq nk_B T \ll nM \simeq \rho$, so $P \simeq 0$. Hence, using the conservation of energy equations and solving the FRW equation we have:

$$\begin{aligned} \rho &\propto a^{-3}, \\ a &\propto t^{2/3}. \end{aligned}$$

If $a(t_0) = 1$ we have $a = (t/t_0)^{2/3}$.

The case of *relativistic matter*, often called *radiation*, encompasses particles which are massless like photons or close to massless like neutrinos. Using the equation of state and conservation of energy, we have that for radiation $P = \rho/3$ so, again, using conservation of energy and the FRW equation we find

$$\begin{aligned} \rho &\propto a^{-4}, \\ a &\propto t^{1/2}. \end{aligned}$$

Normalising the scale factor as above, we have $a = (t/t_0)^{1/2}$. Both matter and radiation dominated universes are *decelerating*.

Finally, we should consider the very special case of a *cosmological constant*, introduced by Einstein to construct a static universe [10]. Such an odd situation arises in the extreme case of $P = -\rho$. You may find that such an equation of state is obeyed by vacuum fluctuations of matter. Such type of matter can be described by the $\Lambda$ we found in Eq. (1). The solutions are straightforward:

$$\begin{aligned} \rho &\propto \text{constant}, \\ a &\propto \exp\left(\sqrt{\frac{\Lambda}{3}}t\right) = \exp(Ht). \end{aligned}$$

We can normalise so that $a = \exp[H(t - t_0)]$. Note a few things: the Hubble constant, $H = \dot{a}/a$ is actually constant, the scale factor is accelerating, and there is no finite time at which $a(t) = 0$ (no 'Big Bang').

So far we have considered one type of matter at a time but it would make more sense to consider a mix. For example, we know that there are photons *and* protons in the Universe, and, it turns out, we should also consider $\Lambda$ (or something like it). So we need to include these different types of energy density in the FRW equations:

$$\left(\frac{\dot{a}}{a}\right)^2 = \frac{8\pi G}{3}\left(\frac{\rho_{M0}}{a^3} + \frac{\rho_{R0}}{a^4}\right) + \frac{\Lambda}{3}.$$

Depending on the evolution of each component as a function of $a$, they will dominate the dynamics of the Universe at different times. In Fig. 2 we plot the energy densities as a function of scale factor and we can clearly see the three stages in the Universe's evolution: a *radiation era* (RD), followed by a *matter era* (MD), ending up with a *cosmological constant era* more commonly known as a $\Lambda$ *era* ($\Lambda$D).

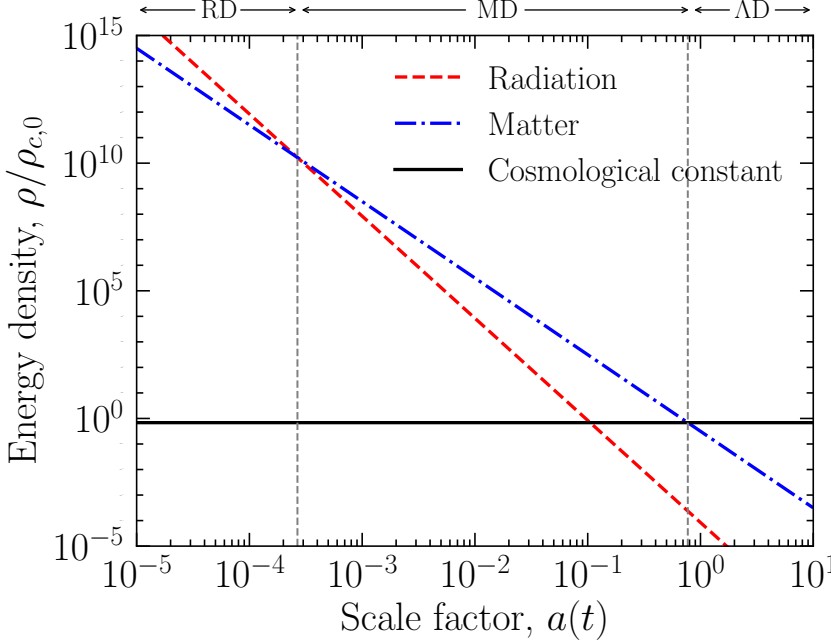

Figure 2: The energy density of radiation, matter and the cosmological constant as a function of scale factor.

What about the curvature of 3-space? We can see that the term proportional to $k$ will only be important at late times, when it dominates over the energy density of dust. In other words, in such a universe (without $\Lambda$) we can say that *curvature dominates* at late times. Let us now consider two possibilities. First of all, let us take $k < 0$. We then have that

$$\left(\frac{\dot{a}}{a}\right)^2 = \frac{8\pi G}{3}\rho + \frac{|k|}{a^2}.$$

When curvature dominates we have that

$$\left(\frac{\dot{a}}{a}\right)^2 = \frac{|k|}{a^2},$$

so $a \propto t$. In this case, the scale factor grows at the speed of light.

We can also consider $k > 0$. From the FRW equations we see that there is a point, when $8\pi G/3\rho = k/a^2$ and therefore $\dot{a} = 0$ when the Universe stops expanding. At this point the Universe starts contracting and evolves to a *Big Crunch*. Clearly geometry is intimately tied to destiny. If we know the geometry of the Universe, we know its future.

It is convenient to define a more compact notation. We can define the *critical energy density*, $\rho_c \equiv 3H_0^2/8\pi G$. If we take $H_0 = 100h$ km s$^{-1}$Mpc$^{-1}$ (where we have introduced the dimensionless parameter $h$ to define the value of the Hubble constant), we have that $\rho_c = 1.9 \times 10^{-26}h^2$ kg m$^{-3}$ which corresponds to a few atoms of Hydrogen per cubic meter. Compare this with the density of water which is $10^3$ kg m$^{-3}$. We can then define the *fractional energy density* or *density parameter*, $\Omega \equiv \rho/\rho_c$. This is the value at $t_0$, i.e. today. We can also define it as a function of time, as $\Omega(a)$, replacing $H_0$ by $H(t)$ in the definition of $\rho_c$. If there are various contributions to the energy density, we can define the fractional energy densities of each one of these contributions. For example, we have $\Omega_R \equiv \rho_R/\rho_c$ and $\Omega_M \equiv \rho_M/\rho_c$. It is also convenient to define two additional $\Omega_i$: $\Omega_\Lambda \equiv \Lambda/3H_0^2$ and $\Omega_k \equiv -k/a^2H_0^2$. We then have $\Omega = \Omega_R + \Omega_M + \Omega_\Lambda$.

We can divide the FRW equation through by $\rho_c$ to find that it can be rewritten as

$$H^2(a) = H_0^2\left[\frac{\Omega_M}{a^3} + \frac{\Omega_R}{a^4} + \frac{\Omega_k}{a^2} + \Omega_\Lambda\right]. \tag{1}$$

This is, in some sense, a complete mathematical model of the Universe. It is an ordinary differential equation, given by

$$\left(\frac{\dot{a}}{a}\right)^2 = H^2\left(a|H_0, \Omega_M, \Omega_R, \Omega_k, \Omega_\Lambda\right),$$

which depends on a number of parameters, $\{H_0, \Omega_M, \Omega_R, \Omega_k, \Omega_\Lambda\}$. The $\Lambda$-CDM model corresponds to one particular choice of parameters. To find these parameters we need to measure certain properties of the Universe which we now turn to.

## 4   Distance measures and redshifts

We can now explore some of the properties of such universes. Pick two objects (galaxies for example) that lie at a given distance from each other and are at rest in given coordinates. Due to the expansion of the Universe, at time $t_1$ they are at a distance $d_1$, while at a time $t_2$ they are at a distance $d_2$. We have that during that time interval, the change between $d_1$ and $d_2$ is given by

$$\frac{d_2}{d_1} = \frac{a(t_2)}{a(t_1)},$$

and, because of the cosmological principle, this is true *whatever* two points we would have chosen. It then makes sense to parametrise the distance between the two points as

$$d(t) = a(t)x,$$

where $x$ is completely independent of $t$. We can see that we have already stumbled upon $x$ when we wrote down the metric for a homogeneous and isotropic space time. It is the set of coordinates $(x^1, x^2, x^3)$ that remain unchanged during the evolution of the Universe. These time-independent coordinates are known as *comoving* or *conformal* coordinates. We recover the real, *physical* coordinates by multiplying the comoving coordinates by the scale factor, $a(t)$.

We can work out how quickly the two objects we considered are moving away from each other. We have that their relative velocity is given by

$$v = \dot{d} = \dot{a}x = \frac{\dot{a}}{a}ax = \frac{\dot{a}}{a}d \equiv Hd.$$

In other words, the recession speed between two objects is proportional to the distance between them. This equality applied today (at $t_0$) is

$$v = H_0 d,$$

and is known as *Hubble's Law*, where $H_0$ is the Hubble constant we found in the previous section, in the Friedmann equation. Objects at rest whose motion is described by this law are said to follow the *Hubble flow*.

Consider a photon with wavelength $\lambda$ being emitted at one point and observed at some other point. We have that the Doppler shift is (let us momentarily re-introduce $c$) given by

$$\lambda' \simeq \lambda(1 + \frac{v}{c}).$$

We can rewrite it in a differential form

$$\frac{d\lambda}{\lambda} \simeq \frac{dv}{c} = \frac{\dot{a}}{a}\frac{dr}{c} = \frac{\dot{a}}{a}dt = \frac{da}{a},$$

and integrate to find $\lambda \propto a$. We therefore have that wavelengths are stretched with the expansion of the Universe. It is convenient to define the factor by which the wavelength is stretched by

$$z = \frac{\lambda_r - \lambda_e}{\lambda_e} \to 1 + z \equiv \frac{a_0}{a},$$

where $a_0$ is the scale factor today (throughout these lecture notes we will choose a convention in which $a_0 = 1$). We call $z$ the *redshift*.

Distances play an important role if we are to map out behaviour of the Universe in detail [11]. We have already been exposed to Hubble's law, from which we can extract Hubble's constant, $H_0$. From Hubble's constant we can define a *Hubble time*

$$t_H = \frac{1}{H_0} = 9.78 \times 10^9 \, h^{-1} \text{ yr},$$

and the *Hubble distance*

$$D_H = \frac{1}{H_0} = 3000 \, h^{-1} \text{ Mpc},$$

recalling that we have set $c = 1$. These quantities set the scale of the Universe and give us a rough idea of how old it is and how far we can see. They are only estimates and to get a firmer idea of distances and ages, we need to work with the metric and FRW equations more carefully.

To actually figure out how far we can see, we need to work out how far a light ray travels over a given period of time. To be specific, what is the comoving distance, $\chi$, to a source, and how is this related to quantities we measure in practice? For a light ray, $ds^2 = 0$, we have that

$$\frac{dr^2}{1 - kr^2} = \frac{dt^2}{a^2(t)}$$

The time integral gives us the *comoving distance*:

$$\chi(t) = \int_t^{t_0} \frac{dt'}{a(t')}.$$

Recall that we have $-k = \Omega_k/D_H^2$. Performing the radial integral (and assuming the observer is at $r = 0$) we have

$$\chi(D) = \int_0^D \frac{dr}{\sqrt{1 - kr^2}} = \begin{cases} \frac{D_H}{\sqrt{\Omega_k}} \sinh^{-1}\left[\sqrt{\Omega_k} D/D_H\right] & \text{for } \Omega_k > 0 \\ D & \text{for } \Omega_k = 0 \\ \frac{D_H}{\sqrt{|\Omega_k|}} \sin^{-1}\left[\sqrt{|\Omega_k|} D/D_H\right] & \text{for } \Omega_k < 0 \end{cases},$$

where $D$ is the radial comoving coordinate distance to the source. We can invert the expression to find

$$D = \begin{cases} \frac{D_H}{\sqrt{\Omega_k}} \sinh\left[\sqrt{\Omega_k} \chi/D_H\right] & \text{for } \Omega_k > 0 \\ \chi & \text{for } \Omega_k = 0 \\ \frac{D_H}{\sqrt{|\Omega_k|}} \sin\left[\sqrt{|\Omega_k|} \chi/D_H\right] & \text{for } \Omega_k < 0 \end{cases}.$$

Suppose now we look at an object of a finite size which is transverse to our line of sight and lies at a certain distance from us. If we divide the physical transverse size of the object by the angle that object subtends in the sky (the angular size of the object) we obtain the *physical angular diameter distance*:

$$D_A = \frac{D}{1 + z}.$$

Hence, if we know that size of an object and its redshift we can work out, for a given Universe, $D_A$. It turns out that one works more often with the *comoving angular diameter distance*, which is simply $D$.

Alternatively, we may know the brightness or luminosity of an object at a given distance. We know that the flux of that object at a distance $D_L$ is given by

$$F = \frac{L}{4\pi D_L^2}.$$

$D_L$ is aptly known as the *luminosity distance* and is related to other distances through:

$$D_L = (1 + z)D = (1 + z)^2 D_A.$$

The comoving, angular and luminosity distances are compared in Fig. 3 for $\Lambda$CDM. Note that the comoving angular diameter distance, $D$, is equal to the comoving distance, $\chi$, since $\Omega_k = 0$ in $\Lambda$CDM (see Eq. (2)). We also point out that the angular distance decreases at large redshifts! It turns out that, in astronomy, one often works with a logarithmic scale, i.e. with *magnitudes*. One can define the *distance modulus*:

$$DM \equiv 5 \log\left(\frac{D_L}{10 \text{ pc}}\right),$$

and it can be measured from the *apparent magnitude m* (related to the flux at the observer) and the *absolute magnitude M* (what it would be if the observer was at 10 pc from the source) through

$$\mu = M + DM.$$

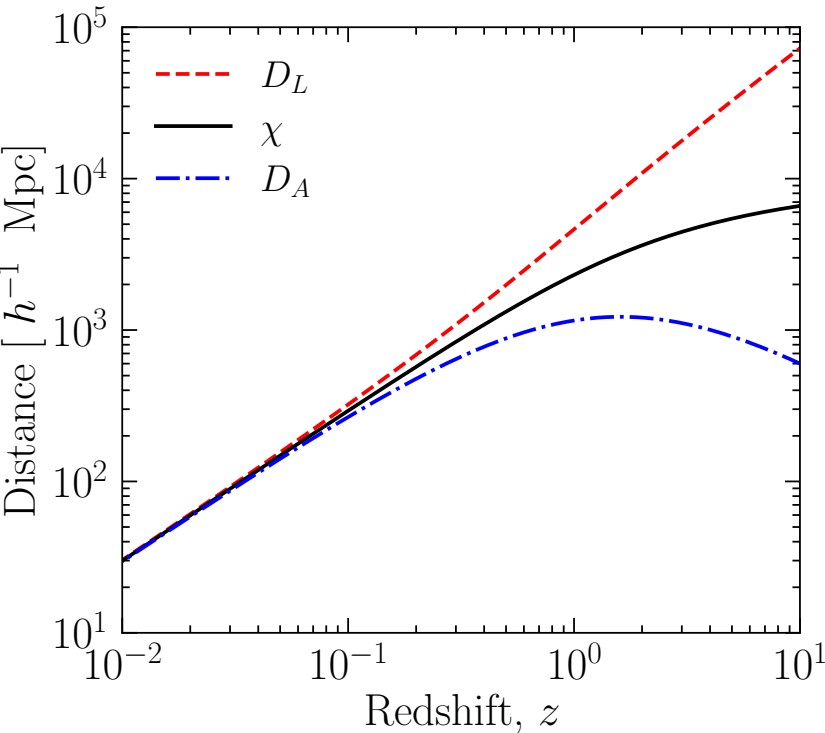

Figure 3: Comparison of three different distance measures in ΛCDM: comoving distance, $\chi$, angular diameter distance, $D_A$, and the luminosity distance, $D_L$.

We now have a plethora of distances which can be deployed in a range of different observations. They clearly depend on the universe we are considering, i.e. on the values of $H_0$, and the various $\Omega_i$. While $\Omega_k$ will dictate the geometry, $\chi$ will depend on how the Universe evolves. It is useful to rewrite $\chi$ in a few different ways. It is useful to transform the time integral in $\chi$ to an integral in $a$:

$$\chi = \int_t^{t_0} \frac{dt'}{a(t')} = \int_a^1 \frac{da}{a^2 H(a)} = D_H \int_a^1 \frac{da}{a^2 \sqrt{\Omega_M/a^3 + \Omega_R/a^4 + \Omega_k/a^2 + \Omega_\Lambda}}.$$

As before, we can think of this as a mathematical model for our universe where distances, are a function of the scale factor $a$, (or redshift, $z$) and a set of parameters, $\{H_0, \Omega_M, \Omega_R, \Omega_k, \Omega_\Lambda\}$. Different choices of parameters will give us different curves for $\chi$, $D_A$ and $D_L$ as a function of $\{H_0, \Omega_M, \Omega_R, \Omega_k, \Omega_\Lambda\}$. The challenge is then to measure $(z, \chi)$, $(z, D_A)$ and/or $(z, D_L)$.

We have been focusing on distances, but our model also allows us to estimate the age of the Universe. We might use the Hubble time to give us a rough estimate of the age of the Universe, but we can we do better if we resort to FRW equations. We have that $\dot{a} = aH$ so

$$dt = \frac{da}{aH} \rightarrow \int_0^{t_0} dt = \int_0^1 \frac{da}{aH} = t_0,$$

which gives us

$$t_0 = \frac{1}{H_0} \int_0^1 \frac{da}{a \sqrt{\Omega_M/a^3 + \Omega_R/a^4 + \Omega_k/a^2 + \Omega_\Lambda}}.$$

We can use the above equation quite easily. For a flat, dust-dominated Universe we find

$t_0 = 2/(3H_0)$. If we now include a cosmological constant as well, we find

$$t_0 = H_0^{-1} \int_0^1 \frac{da}{a\sqrt{\Omega_M/a^3 + \Omega_\Lambda}}.$$

At $\Omega_\Lambda = 0$ we simply retrieve the matter dominated result, but the larger $\Omega_\Lambda$ is, the older the Universe.

Finally, let us revisit Hubble's law. We worked out the relationship between velocities and distance for two objects which were very close to each other. If we want to consider objects which are further apart (not too distant galaxies), we can Taylor expand the scale factor today, we find that

$$a(t) = a(t_0) + \dot{a}(t_0)[t - t_0] + \frac{1}{2}\ddot{a}(t_0)[t - t_0]^2 + \cdots.$$

Reintroducing $c$ again, assume that the distance to the emitter at time $t$ is roughly given by $d \simeq c(t_0 - t)$. We can then rewrite it as

$$(1 + z)^{-1} = 1 - H_0 \frac{d}{c} - \frac{q_0 H_0^2}{2}\left(\frac{d}{c}\right)^2 + \cdots.$$

For $q_0 = 0$ and small $z$ we recover the Hubble law, $cz = H_0 d$. As we go to higher redshift, this is manifestly not good enough and we can then measure $q_0$. But recall the Raychaudhuri equation for this Universe:

$$\frac{\ddot{a}}{a} = -\frac{4\pi G}{3}\rho + \frac{\Lambda}{3}.$$

Divide by $H_0^2$ and we have that the *deceleration parameter*

$$q_0 \equiv -\frac{a(t_0)\ddot{a}(t_0)}{\dot{a}^2(t_0)} = \frac{1}{2}\Omega_M - \Omega_\Lambda.$$

Which means that, by going beyond the linear Hubble law, we can measure $\Omega_M$ and $\Omega_\Lambda$.

Note that we have kept the model simple but what if we loosened up what $\Lambda$ is? Let us call it *dark energy* and characterise it in terms of a general $w(a)$. A general $w(a)$ is quite a lot to add the model: so far we have only added numbers as parameters but now we are adding a function of $a$ which has, effectively, an infinite number of parameters. We can consider the simplest, non-trivial, parametrisation which allows for some time evolution

$$w(a) \simeq w_0 + w_a(1 - a).$$

This is often referred to as the *Chevallier-Polarski-Linder* (CPL) parametrisation [12, 13]. Solving for the corresponding energy density, we have

$$\rho_{\text{DE}} \propto \frac{e^{3w_a(a-1)}}{a^{3(1+w_0+w_a)}}$$

and we now have

$$\chi = D_H \int_a^1 \frac{da}{a^2\sqrt{\Omega_M/a^3 + \Omega_R/a^4 + \Omega_k/a^2 + \Omega_{\text{DE}}e^{3w_a(a-1)}/a^{3(1+w_0+w_a)}}}.$$

This means we have added two parameters to our model $w_0, w_a$ as well as the others, $\{H_0, \Omega_M, \Omega_R, \Omega_k, \Omega_\Lambda\}$.

Yet another possibility is that we assume a microphysical model for the dark energy such as, for example, that it is a *quintessence* field. This is a scalar field, $\varphi$, with an energy-momentum tensor given by

$$T_{\mu\nu} = \partial_\mu \varphi \partial_\nu \varphi + \frac{1}{2} g_{\mu\nu} [\partial^\alpha \varphi \partial_\alpha \varphi + V(\varphi)].$$

Assuming the cosmological principle we have the evolution equation

$$\ddot{\varphi} + 3H\dot{\varphi} + dV/d\varphi = 0,$$

and the field's energy density and pressure are given by

$$
\begin{aligned}
\rho_\varphi &= \frac{1}{2}\dot{\phi}^2 + V, \\
P_\varphi &= \frac{1}{2}\dot{\phi}^2 - V.
\end{aligned}
$$

If the potential energy dominates, we have that $w_\varphi < 0$, but if the $\dot{\varphi} \neq 0$, it can evolve. Consider the case where $V = \frac{1}{2}m^2\varphi^2$ and start the scalar field away from the minimum, at $\varphi_0$ and with $\dot{\varphi} \simeq 0$. Then, initially, $w_\varphi \simeq -1$, but as it evolves, we will have that $\varphi \sim 1/a^{3/2} \times \cos(mt)$. For $mt \gg 1$, we have that $\langle P_\varphi \rangle \simeq 0$ (where the angle brackets imply a time average over a number of oscillations) and $\rho_\varphi \sim 1/a^3$, i.e. it evolves as dark matter ("axion" dark matter) [14]. There is a hope that we may be able to, with cosmological data, constrain and figure out the properties of dark energy.

# 5 Thermal equilibrium and recombination

We now look at how the contents of the Universe are affected by expansion. The first property which we must consider is that as the Universe expands, its contents cool down. How can we see that? Let us focus on the radiation contained in the Universe. In the previous sections we found that the energy density in radiation decreases as $\rho \propto 1/a^4$.

What else can we say about radiation? Let us make a simplifying assumption, that it is in thermal equilibrium and thus behaves like a blackbody. For this to be true, the radiation must interact very efficiently with itself to redistribute any fluctuations in energy and occupy the maximum entropy state. It can be described in terms of an *occupation number per mode* given by

$$F(\nu) = \frac{2}{\exp\frac{h\nu}{k_B T} - 1},$$

where $\nu$ is the frequency of the photon. This corresponds to an *energy density per mode*

$$\epsilon(\nu)d\nu = \frac{8\pi h\nu^3 d\nu}{\exp\frac{h\nu}{k_B T} - 1}.$$

If we integrate over all frequencies we have that the energy density in photons is:

$$\rho_\gamma = \frac{\pi^2}{15}(k_B T)\left(\frac{k_B T}{\hbar}\right)^3.$$

We have therefore that $\rho_\gamma \propto T^4$. Hence if radiation is in thermal equilibrium we have that $T \propto \frac{1}{a}$.

Is this the temperature of the Universe? Two ingredients are necessary. First of all, everything else has to feel that temperature, which means that they have to interact (even if only indirectly) with the photons. For example, the scattering of electrons and protons is through the emission and absorption of photons. And once again, at sufficiently high temperatures, everything interacts quite strongly.

Another essential ingredient is that the radiation must dominate over the remaining forms of matter in the Universe. We have to be careful with this because we know that different types of energy will evolve in different ways as the Universe expands. For example we have that the energy density of dust (or non-relativistic matter) evolves as $\rho_{NR} \propto a^{-3}$ as compared to $\rho_\gamma \propto a^{-4}$ so even if $\rho_\gamma$ was dominant at early times it may be negligible today. However we also know that the *number density* of photons, $n_\gamma \propto a^{-3}$ as does the number density of non-relativistic particles, $n_{NR} \propto a^{-3}$. If we add up all the non-relativistic particles in the form of neutrons and protons (which we call baryons), we find that number density of baryons, $n_B$ is very small compared to the number density of photons. In fact, we can define the *baryon to entropy ratio*, $\eta_B \equiv n_B/n_\gamma \simeq 10^{-10}$. As we can see, there are many more photons in the Universe than particles like protons and neutrons. So it is safe to say that the temperature of the photons sets the temperature of the Universe.

We can think of the Universe as a gigantic heat bath which is cooling with time. The temperature decreases as the inverse of the scale factor. To study the evolution of matter in the Universe we must now use statistical mechanics to follow the evolution of the various components as the temperature decreases. An ideal gas of bosons or fermions has an occupation number per mode (now labeled in terms of momentum $\mathbf{p}$) given by

$$F(\mathbf{p}) = \frac{g}{\exp\left(\frac{E-\mu}{k_B T}\right) \pm 1},$$

where $g$ is the degeneracy factor, $E = \sqrt{p^2 + M^2}$ is the energy, $\mu$ is the chemical potential and + (-) corresponds to the Fermi-Dirac (Bose-Einstein) distribution. We can use this expression to calculate some macroscopic quantities such as, for example, the number density

$$n = \frac{g}{h^3} \int \frac{d^3 p}{\exp\left(\frac{E-\mu}{k_B T}\right) \pm 1}.$$

It is instructive to consider two limits. First of all let us take the case where the temperature of the Universe corresponds to energies which are much larger than the rest mass of the individual particles, i.e. $k_B T \gg M$ and let us take $\mu \simeq 0$. We then have that the energy density is given by

$$\rho c^2 = g \frac{\pi^2}{30} (k_B T) \left(\frac{k_B T}{\hbar}\right)^3 \quad \text{(B.E.)}$$

$$\rho c^2 = \frac{7}{8} g \frac{\pi^2}{30} (k_B T) \left(\frac{k_B T}{\hbar}\right)^3 \quad \text{(F.D.)}$$

and pressure satisfies $P = \rho/3$. As you can see these are the properties of radiation. In other words, even massive particles will behave like radiation at sufficiently high temperatures. At low temperatures we have $k_B T \ll M$ and for both fermions and bosons the macroscopic quantities are given by:

$$n = g \left(\frac{2\pi}{h^2}\right)^{\frac{3}{2}} (M k_B T)^{3/2} \exp\left(-\frac{M}{k_B T}\right),$$

$$\rho c^2 = Mn,$$

$$P = n k_B T \ll Mn = \rho.$$

This last expression tells us that the pressure is negligible as it should be for non-relativistic matter.

This calculation has already given us an insight into how matter evolves during expansion. At sufficiently early times it all looks like radiation. As it cools down and the temperature falls below mass thresholds, the number of particles behaving relativistically decreases until when we get to today, there are effectively only a few types of particles which behave relativistically: neutrinos and photons. We denote the *effective number of relativistic degrees of freedom* by $g_*$ and the energy density in relativistic degrees of freedom is given by

$$\rho \;=\; g_* \frac{\pi^2}{30}(k_B T)\left(\frac{k_B T}{\hbar}\right)^3.$$

Until now we have considered things evolving passively, subjected to the expansion of the Universe. But we know that the interactions between different components of matter can be far more complex. Let us focus on the realm of chemistry, in particular on the interaction between one electron and one proton. From atomic physics and quantum mechanics you already know that an electron and a proton may bind together to form a Hydrogen atom. To tear the electron away we need an energy of about 13.6 eV. But imagine now that the universe is sufficiently hot that there are particles zipping around that can knock the electron out of the atom. We can imagine that at high temperatures it will be very difficult to keep electrons and protons bound together. If the temperature of the Universe is such that $T \simeq 13.6$ eV, then we can imagine that there will be a transition between ionised and neutral hydrogen.

We can work this out in more detail (although not completely accurately) if we assume that this transition occurs in thermal equilibrium throughout. Let us go through the steps that lead to the *Saha equation* [15]. Assume we have an equilibrium distribution of protons, electrons and hydrogen atoms. Let $n_p$, $n_e$ and $n_H$ be their number densities. In thermal equilibrium (with $T \ll M$) we have that the number densities are given by

$$n_i = g_i\left(\frac{2\pi}{h^2}\right)^{\frac{3}{2}}(M_i k_B T)^{\frac{3}{2}}\exp\frac{\mu_i - M_i}{k_B T},$$

where $i = p, n, H$. In chemical equilibrium we have that $\mu_p + \mu_e = \mu_H$, so that

$$n_H = g_H\left(\frac{2\pi}{h^2}\right)^{\frac{3}{2}}(M_H k_B T)^{\frac{3}{2}}\exp\frac{-M_H}{k_B T}\exp\frac{(\mu_p + \mu_e)}{k_B T}.$$

We can use the expressions for $n_p$ and $n_e$ to eliminate the chemical potentials and obtain an expression for the ionisation fraction, $X \equiv n_p/n_B$, such that

$$\frac{1-X}{X^2} \simeq 3.8\eta_B\left(\frac{k_B T}{M_e}\right)^{\frac{3}{2}}\exp\frac{B}{k_B T},$$

where we have assumed that i) $M_p \simeq M_H$, ii) the binding energy is $B \equiv -M_H + M_p + M_e = 13.6$ eV, iii) $n_B = n_p + n_H$ iv) $n_e = n_p$ and finally $g_p = g_e = 2$ and $g_H = 4$. Finally we have that we are in thermal equilibrium so we have an expression for $n_\gamma$ and we get

$$\frac{1-X}{X^2} \simeq 3.8\eta_B\left(\frac{k_B T}{M_e}\right)^{3/2}\exp\left(\frac{B}{k_B T}\right). \tag{2}$$

This is the Saha equation. It tells us how the ionisation fraction, $X$, evolves as a function of time. We can see its evolution in Fig 4. At sufficiently early times we will find that $X = 1$, i.e. the Universe is completely ionised. As it crosses a certain threshold, electrons and protons combine to form Hydrogen. This happens when the temperature of the Universe is $T \sim 3000K$

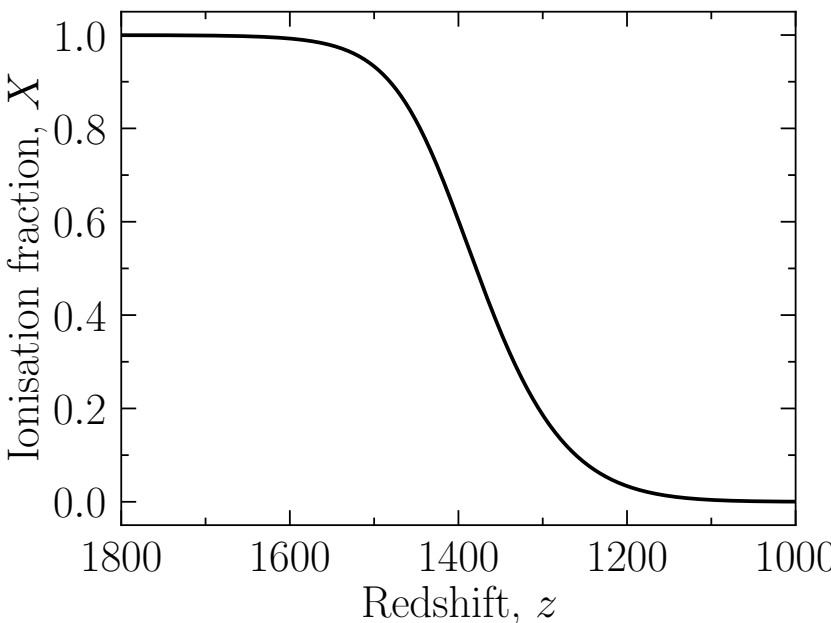

Figure 4: The evolution of the ionisation fraction, $X$, as a function of redshift, $z$, predicted by the Saha equation Eq. (2).

or 0.308 eV, i.e. when it was approximately $380,000$ years old, at a redshift of $z \simeq 1100$. We would naively expect this to happen at 13.6 eV, but the prefactors in front of the exponential play an important role. One way to think about it is that, at a given temperature there will always be a few photons with energies larger than the average temperature. Thus energetic photons only become unimportant at sufficiently low temperatures.

What does this radiation look like to us? At very early times, before recombination, this radiation will be in thermal equilibrium and satisfy the Planck spectrum, given above. After recombination, the electrons and protons combine to form neutral hydrogen and the photons will be left to propagate freely. The only effect will be the redshifting due to the expansion. The net effect is that the shape of the spectrum remains the same, the peak shifting as $T \propto 1/a$. So even though the photons are not in thermal equilibrium anymore, the spectrum will still be that of thermal equilibrium, with a temperature $T_0 = 2.75$ Kelvin.

The history of each individual photon can also be easily described. Let's work backwards. After recombination, a photon does not interact with anything and simply propagates forward at the speed of light. Its path will be a straight line starting off at the time of recombination and ending today. Before recombination, photons are highly interacting with a very dense medium of charged particles, the protons and electrons. This means that they are constantly scattering off particles, performing something akin to a random walk with a very small step length. For all intents and purposes, they are glued to the spot unable to move. So one can think of such a photon's history as starting off stuck at some point in space and, at recombination, being released to propagate forward until now.

We can take this even further. If we look from a specific observing point (such as the Earth or a satellite), we will be receiving photons from all directions that have been travelling in a straight line since the Universe recombined. All these straight lines will have started off at the same time and at the same distance from us – i.e. they will have started off from the surface of a sphere. This surface, known as the *surface of last scattering*, is what we see when we look at the relic radiation. It is very much like a photograph of the Universe when it was 380,000 years old.

# 6 Newtonian perturbation theory

Our assumption of homogeneity and isotropy is borne out by our observations of the cosmic microwave background, which we find to be isotropic to within one part in a hundred thousand. Yet we know that the observable universe is remarkably smooth and isotropic but it is not perfectly so. We see a plethora of structures, from clusters, filaments and walls of galaxies to large empty voids that can span hundreds of millions of light years. Indeed, the fact that there are galaxies, stars and planets indicate that the Universe is not at all smooth as we observe it on smaller and smaller scales. Hence to have a complete understanding of the dynamics and state of the Universe and to be able to accurately predict its large scale structure, we must go beyond describing it just in terms of a scale factor, overall temperature, density and pressure.

If we are to explore departures from homogeneity, we must study the evolution of the energy density, $\rho$, pressure, $P$ and gravity in an expanding universe in a more general context, allowing for spatial variations in these various contexts. We will restrict ourselves to Newtonian gravity, encapsulated in the Newtonian potential, $\Phi$, which will give us the qualitative behaviour of perturbations that we might find in a proper, general relativistic treatment. This is a good approximation on sufficiently small scales but where gravity is still weak.

Let us focus on the evolution of pressureless matter, appropriate for the case of massive, non-relativistic particles. The evolution of a gravitating pressureless fluid is governed by a set of conservation equations known as the Euler equations. We have that *conservation of energy* is given by

$$\frac{\partial \rho}{\partial t} + \nabla \cdot (\rho \vec{V}) = 0,$$

while *conservation of momentum* is given by

$$\frac{\partial V}{\partial t} + (\vec{V} \cdot \nabla)\vec{V} = -\nabla \Phi - \frac{1}{\rho}\nabla P.$$

Note that we have had to introduce the fluid velocity, $\vec{V}$ into our system. These conservation equations are complemented by the Newton-Poisson equation

$$\nabla^2 \Phi = 4\pi G \rho,$$

which tell us how this system behaves under gravity.

We can clearly see that this set of equations are strictly valid for a universe dominated by pressureless matter if we attempt to solve for the mean density $\rho_0$ and mean expansion $\vec{V}_0 = H\vec{r}$ corresponding to a homogeneous and isotropic universe. Solving the conservation of energy equation we have that

$$\frac{\partial \rho}{\partial t} = -\nabla \cdot (\rho_0 \vec{V}_0) = -\rho \nabla \cdot \vec{V}_0 = -3H\rho,$$

which gives us $\rho \propto a^{-3}$.

The Euler equations are, in general, difficult to solve. We can, however, study what happens when the Universe is mildly inhomogeneous, i.e. we can consider small perturbations around their mean (background) values so that the total density, pressure, velocity and gravitational potential at a given point in space can be written as $\rho = \rho_0 + \delta\rho$, $P = P_0 + \delta P$, $\vec{V} = \vec{V}_0 + \delta\vec{v}$, $\Phi = \Phi_0 + \delta\Phi$, where $\delta\rho/\rho \ll 1$, $\delta P/P \ll 1$ and so on. This approach is known as *cosmological perturbation theory* – it involves the study of small perturbations to a FRW universe and we will find that the evolution equations greatly simplify in this regime. We can first start off with the conservation of energy equation

$$\frac{\partial(\rho_0 + \delta\rho)}{\partial t} + \nabla \cdot [(\rho_0 + \delta\rho)(\vec{V}_0 + \delta\vec{v})] = 0,$$

which we can expand to give us:

$$\frac{\partial \rho_0}{\partial t} + \nabla \cdot (\rho_0 \vec{V}_0) + \frac{\partial \delta \rho}{\partial t} + \nabla \cdot (\rho_0 \delta \vec{v}) + \nabla \cdot (\delta \rho \vec{V}_0) + \nabla \cdot (\delta \rho \delta \vec{v}) = 0.$$

The first two terms satisfy the conservation equations as seen above while the last term is a product of two very small quantities and hence is negligible. It is possible to further simplify the equations using $\nabla \cdot \vec{V}_0 = 3H$ and defining $\delta \equiv \delta \rho / \rho_0$. If we convert the partial derivative in time to a total time derivative

$$\frac{d\delta}{dt} = \frac{\partial \delta}{\partial t} + \vec{V}_0 \cdot \nabla \delta,$$

we then find that the first order conservation of energy equation reduces to

$$\frac{d\delta}{dt} + \nabla \cdot \delta \vec{v} = 0.$$

The same can be done to the conservation of momentum equation,

$$\frac{d\delta \vec{v}}{dt} + H \delta \vec{v} \ = \ -c_S^2 \nabla \delta - \nabla \delta \Phi,$$

where we have defined the speed of sound of this fluid to be $c_s^2 = \frac{\nabla \delta P}{\nabla \delta \rho}$ and the perturbed Newton-Poisson equation[1] becomes

$$\nabla^2 \Phi = 4\pi G \rho_0 \delta.$$

The system has now been simplified to a set of linear differential equations with time dependent coefficients, which can be solved either numerically or approximately using Fourier transforms.

There is a further transformation we can do to simplify the system. First of all it is important to note that we have been working in physical coordinates, $\vec{r}$, and that it is much more convenient to switch to conformal coordinates, $\vec{x}$ (i.e. coordinates that are defined on the space-time grid); we then have $\vec{r} = a\vec{x}$ so that gradients between the two coordinate systems are related through $\nabla_r = \frac{1}{a}\nabla_x$ and the velocity perturbations are related through $\delta \vec{v} = a\vec{u}$. If we make a further simplifying assumption that there are no vortical flows in the fluid, we can define a new variable $\Theta = \nabla \cdot \vec{u}$. We then have that perturbed conservation of momentum equation becomes

$$\dot{\Theta} + 2H\Theta = -\frac{c_s^2}{a^2}\nabla^2\delta - 4\pi G \rho_0 \delta.$$

Combined with perturbed conservation of mass equation we can rewrite the perturbed Euler equations as a second order linear partial differential equation $\delta$:

$$\ddot{\delta} + 2H\dot{\delta} - \frac{c_s^2}{a^2}\nabla^2\delta = 4\pi G \rho_0 \delta.$$

If we take the Fourier transform[2], $\delta \to \delta_{\mathbf{k}}$, we have that

$$\ddot{\delta}_k + 2H\dot{\delta}_k = \left( -\frac{c_s^2}{a^2}k^2 + 4\pi G \rho_0 \right)\delta.$$

---

[1]In a homogeneous and isotropic Universe we can assume that $\Phi_0 = 0$.

[2]The Fourier transform is taken to be

$$\delta(t, \vec{x}) = \frac{1}{(2\pi)^3} \int d^3k \delta_{\mathbf{k}} \exp(-i\vec{k} \cdot \vec{x}).$$

A cursory glance allows us to identify a number of features in the evolution of $\delta$ without actually solving the system. For a start, it is quite clearly the equation for a damped harmonic oscillator with time dependent damping coefficient and spring constant. The damping is due to the expansion of the Universe and will tend to suppress growth. The spring constant will change sign depending on whether $k$ is large or small. If the positive part of the spring constant, $c_s^2 k^2/a^2$, dominates then we should expect oscillatory behaviour in the form of acoustic waves in the fluid. If the negative term, $4\pi G\rho_0$ dominates, then the evolution will be unstable and we should expect $\delta$ to grow. The *physical* (as opposed to conformal) wavelength, $\lambda_J$, that defines the transition between these two behaviours is given by

$$\lambda_J = c_s \left( \frac{\pi}{G\rho_0} \right)^{\frac{1}{2}}$$

and is known as the *Jeans wavelength*. For $\lambda > \lambda_J$ gravitational collapse dominates and perturbations grow. For $\lambda < \lambda_J$ pressure will win out and perturbations will not grow. We can have a rough idea of how a given system of particles will behave if we note that $c_s^2 \sim (K_B T)/(M)$, where $T$ is the temperature of the system and $M$ is the mass of the individual particles. We can then rewrite the Jeans length as

$$\lambda_J = \left( \frac{\pi K_B T}{GM\rho_0} \right)^{1/2} .$$

It is clear that a hot system or made up of light particles will have a large $\lambda_J$; a cold system with heavy particles will have a small $\lambda_J$.

It is often convenient to write the evolution equation for density perturbations in terms of conformal time $\tau$; recall that $dt = a\,d\tau$, so we can solve for $\tau = \int \frac{dt}{a}$ and we now denote $X' = \frac{dX}{d\tau}$. We obtain

$$\delta_k'' + \frac{a'}{a}\delta_k' + \left( c_s^2 k^2 - 4\pi G\rho a^2 \right)\delta_k = 0.$$

We have focussed on the specific case of a pressureless fluid in the matter dominated era. In this situation we have that $c_s^2 \simeq 0$ and hence $\lambda \gg \lambda_J$. We can therefore discard the term which depends on pressure to get:

$$\ddot{\delta} + 2H\dot{\delta} - \frac{3}{2}H^2\delta = 0,$$

where we have used $\frac{3}{2}H^2 = 4\pi G\rho_0$. From $a = (t/t_0)^{2/3}$, we have $H = 2/3t$ and the solutions are then

$$\delta = C_1 t^{\frac{2}{3}} + C_2 t^{-1}.$$

The second term decays and becomes subdominant very fast and we are left with the first term which can be rewritten as $\delta \sim a$. If we repeat the same calculation now using conformal time we find $\delta_k \propto \eta^2$ and $\eta^{-3}$. Hence we find that in this situation, perturbations grow due to the effect of gravity; the growing solution, $\delta_k \sim a$ is normally called the *growing mode*.

# 7  Relativistic cosmological perturbation theory

In the previous section we used Newtonian gravity to find an equation that describes how over or under densities in the mass distribution evolve. We know, however, that the correct theory

of gravity is general relativity and it would make sense to derive a set of equations for how inhomogeneities evolve which is completely consistent with how we derived the evolution of the Universe as a whole. More generally, we also need relativity to understand the largest scales. Again, we shall apply the rules of linear perturbation theory but now we do so to the Einstein field equations and the relativistic conservation of energy and momentum [1, 2]. We won't push the calculation to its bitter end but will give you a flavour of all the steps involved.

We need to perturb the metric so that

$$ds^2 = \left(g^{(0)}_{\mu\nu} + \delta g_{\mu\nu}\right)dx^\mu dx^\nu.$$

It is useful to work with conformal or comoving time, $dt = a d\tau$. We then assume (for a flat universe):

$$g^{(0)}_{\mu\nu} = a^2(\tau)\eta_{\mu\nu},$$

where $\eta_{\mu\nu}$ is the Minkowski metric. The most general perturbation is

$$ds^2 = a^2(\tau)\left[-(1+2A)d\tau^2 - 2B_i dx^i d\tau + \left(\delta_{ij} + h_{ij}\right)dx^i dx^j\right].$$

In some sense we are reinstating all the freedom we had before (10 functions of $x^\alpha$). But there is a symmetry in general relativity (general coordinate invariance) that, at the linearised level can be thought of as a gauge symmetry,

$$\delta g_{\mu\nu} \to \delta g_{\mu\nu} + \partial_\mu \xi_\nu + \partial_\nu \xi_\mu,$$

for an arbitrary infinitesimal coordinate transformation,

$$x^\mu \to x^\mu + \xi^\mu(x^\alpha).$$

This reduces the functions we need to: 2 scalars, 2 vectors (the components of a vortical field) and to 2 tensors (or gravitational waves).

From now on we will focus on scalars. We can choose

$$ds^2 = a^2(\tau)\left[-(1+2\Psi)d\tau^2 + (1-2\Phi)d\vec{x}^2\right].$$

This means we now need to worry about *two* gravitational potentials. We need to do the same with the energy momentum tensor, expanding it in terms of the perturbations of $\rho$, $P$ and the velocity $U^\mu$.

Working out the linearly perturbed Einstein field equations which (ignoring the cosmological constant for now) will look schematically like

$$\delta G_{\alpha\beta} = 8\pi G \delta T_{\alpha\beta}.$$

Let us focus on the 00 component. In Fourier space, we have that

$$k^2\Phi + 3\mathcal{H}(\Phi' + \mathcal{H}\Psi) = -4\pi G a^2 \delta\rho,$$

where $\mathcal{H} = a'/a$. This equation should look mildly familiar: it looks like the Newton-Poisson equation with some time dependent corrections.

We can use the same process as we used above to show that

$$\delta G^i_{\ j} = A\delta^i_j + \frac{k_i k_j(\Phi - \Psi)}{a^2},$$

where $A$ is a long expression built up out of $\Phi$, $\Psi$ and their derivatives. If we perturb $T^i{}_j$ we find that

$$\delta T^i{}_j = \delta P \delta^i{}_j.$$

The latter is true because we are only perturbing $\rho$, $P$ and $U^\mu$. Now take the perturbed Einstein field equation

$$\delta G^i{}_j = 8\pi G \delta T^i{}_j$$

and try to extract the traceless part. We can do that by applying the projection operator,

$$\mathcal{P}^i{}_j = k^i k_j - \frac{1}{3} k^2 \delta^i{}_j.$$

You quickly find that $\mathcal{P}^i{}_j$ contracted with $\delta^i{}_j$ is zero to leave

$$\frac{2k^2}{3a^2}(\Phi - \Psi) = 0.$$

That is, the Einstein Field equations in the absence of anisotropic stresses set $\Phi = \Psi$. If we now plug that back into the 00 equation we find

$$k^2 \Phi + 3\mathcal{H}(\Phi' + \mathcal{H}\Phi) = -4\pi G a^2 \delta\rho,$$

which, for $a = 1$, gives us the Newton-Poisson equation. If we assume $\Phi' \simeq \mathcal{H}\Phi$ and recall that $\mathcal{H} \simeq 1/\eta$ we have that the Newton Poisson equation is a good approximation when $k\eta \gg 1$. It is useful to introduce a new terminology: we say that modes are *sub-horizon* when $k\tau \gg 1$ and are *super-horizon* when $k\tau \ll 1$. We recover the Newton-Poisson equation when modes are sub-horizon.

The 'relativistic' Newton-Poisson equation can be written in another form if we take the perturbed 0$i$ equations. Taking the perturbed velocity 4-vector, $U^\mu = (1, \vec{v})$ and defining $\theta = \vec{\nabla} \cdot \vec{v}$, we can combine the 00 and 0$i$ components of the Einstein field equation so that

$$\frac{1}{2}\left[G^0{}_0 - i\frac{3\mathcal{H}}{k^2} k^i \delta G^0{}_i\right] = -k^2 \Phi = 4\pi G a^2 \rho_0 \left(\delta - \frac{3\mathcal{H}}{k^2}\theta\right) \equiv 4\pi G a^2 \rho_0 \Delta,$$

where we have defined the *comoving density contrast*, $\Delta$. Again, note the correction to the Newton-Poisson equation. On subhorizon scales, $\Delta \simeq \delta$.

If we now take the trace of the spatial part of the Einstein Field equations we end up with

$$\Phi'' + 3\mathcal{H}\Phi' + \left(2\mathcal{H}' + \mathcal{H}^2\right)\Phi = -4\pi G a^2 \delta P.$$

To close the system we now need to use the relativistic equivalent of energy conservation and the Euler equation. This will come from taking the linear term of the covariant conservation of the energy momentum tensor, $\delta(\nabla_\mu T^{\mu\nu}) = 0$. If we define the sound speed, $c_s^2 = \delta P / \delta\rho$, and recall that the equation of state is $P = w\rho$, we have that the equations are

$$\delta' = -(1 + w)(\theta - 3\Phi') - 3\mathcal{H}^2(c_s^2 - w)\delta,$$
$$\theta' = -\mathcal{H}(1 - 3w)\theta + \frac{c_s^2 k^2}{1 + w}\delta + k^2 \Phi.$$

All together, these equation can be used to study the evolution of perturbations on all scales for a range of $w$ and $c_s^2$ and $\mathcal{H}$.

# 8 The evolution of large scale structure

We now have equations with which we can study how perturbations evolve in time [1,2]. This is a set of ordinary differential equations with time dependent coefficients. We can combine our equations to find one master equation for $\Phi$

$$\Phi'' + 3(1+w)\mathcal{H}\Phi' + wk^2\Phi = 0,$$

which is valid on all scales. On super-horizon scales we can discard the $k^2\Phi$ term and find that $\Phi \sim$ constant in each of the eras (i.e. in which $w =$ constant). This is not the case in the transition between eras and one can show that $\Phi_{\mathrm{MD}} = \frac{9}{10}\Phi_{\mathrm{RD}}$, where RD (MD) stands for Radiation Domination (Matter Domination). This can be seen for the $k = 0.001$ Mpc$^{-1}$ mode in Fig. 5. On sub-horizon scales, in the RD era, we have that

$$\Phi \propto \frac{\sin(x) - x\cos(x)}{x^3},$$

with $x = k\tau/\sqrt{3}$. In the MD era we have, again, that $\Phi \sim$ constant. Clearly equality between the MD and RD eras plays an important role and we can define an important scale, $k_{\mathrm{eq}} = \mathcal{H}_{\mathrm{eq}}$. If we follow a mode from some initial time until now it will look qualitatively different if $k$ is greater or less than $k_{\mathrm{eq}}$ as can be seen in Fig.5.

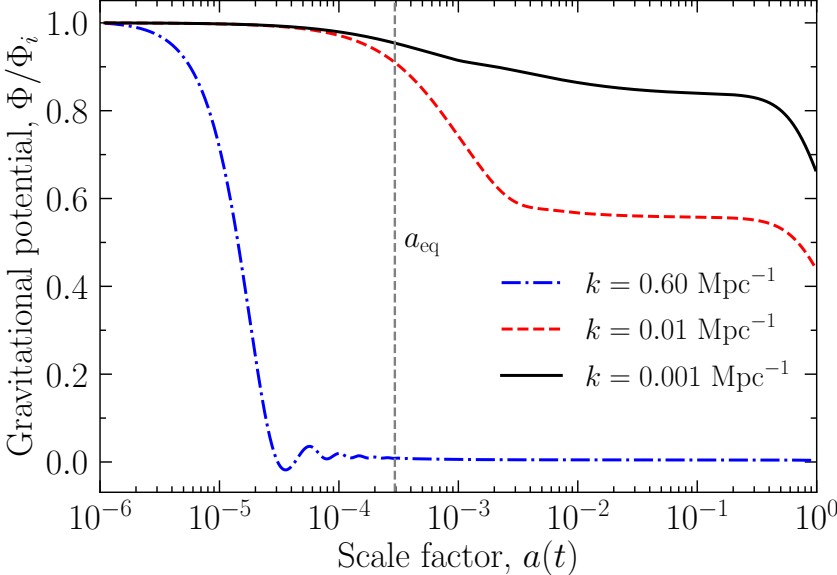

Figure 5: The evolution of the gravitational potential in $\Lambda$CDM for different values of $k$.

We are interested in the distribution of matter and we do so by tracing the behaviour of the comoving matter density contrast (defined above), $\Delta_{\mathrm{M}}$; on subhorizon scales it will map onto $\delta_{\mathrm{M}}$. The two missing regimes are, in the RD era, we have that $\Delta_{\mathrm{M}} \sim \ln a$ on sub-horizon scales, and in a $\Lambda$ dominated era, we have $\delta \sim$ constant. We can summarise the behaviour in the following table

We have simplified somewhat. When we are talking about matter, we are considering non-relativistic matter which include baryons. Baryons interact strongly with the radiation through Thomson scattering, with an interaction strength that depends on $\sigma_T a X n_B$, where $\sigma_T$ is the Thomson cross section, $X$ is the ionisation fraction and $n_B$ is the Baryon number density. As we

| $\Delta_{\mathrm{M}}$ | RD | MD | $\Lambda$ |
|---|---|---|---|
| Super-Horizon | $a^2$ | $a$ | constant |
| Sub-Horizon | $\ln a$ | $a$ | constant |

Table 1: Different regimes in the evolution of the comoving matter density contrast, $\Delta_{\mathrm{M}}$

saw with the Saha equation, before recombination, $X = 1$ and the baryons are tightly coupled with the radiation. We then have that

$$\Delta_{\mathrm{B}} \sim \Delta_{\mathrm{R}} \sim \left(\frac{k}{\mathcal{H}}\right)^2 \Psi \sim (k\tau)^2 \cos\left(\frac{k\tau}{\sqrt{3}}\right).$$

After equality and recombination, $\Delta_{\mathrm{B}} \sim a$, but it will have imprinted on it the oscillations it underwent during the radiation era. The scale of the oscillations will be a function of the sound horizon at the time that baryons decoupled from the photons, $r_d$. The final, total matter density contrast will be the weighted sum of the dark matter density contrast and the baryon density contrast and will, thus, carry the vestiges of these oscillations.

There are two further situations we should examine which do not fit exactly into the formalism we have been using. These occur when there is imperfect coupling between different fluid elements or when the system cannot be described purely in terms of a density field and one must resort to a distribution function. For a brief period during recombination, the mean free path of photons will not be negligible nor will it be infinite. If your recall from our derivation of recombination, the ionisation fraction plummets at around $z \simeq 1100$, the change occurring during a $\Delta z \simeq 80$, During that period, the mean free path of photons will be finite and, because the photons and electrons aren't perfectly coupled, the photons will be able to random walk out of overdensities as they scatter off free electrons. In doing so they will shift matter from over densities to underdensities and damp out perturbations on small scales. This damping scale is known as the Silk damping scale [16].

Massive neutrinos cannot be described as a fluid – they do not interact with each other and their evolution must be studied using the Boltzmann equation. On very large scales they will tend to cluster just like matter and radiation but on small scales, they will tend to *free-stream* i.e. move relativistically from one region of space to another [17]. This will lead to an overall damping effect, wiping out structure on small scales. The damping scale will depend on their mass and is roughly given by

$$\lambda_{FS} \simeq 40 \left(\frac{30 \text{ eV}}{M_\nu}\right) \text{Mpc}.$$

## 9   Initial conditions and random fields

Thus far we have studied the evolution of structure in a variety of scenarios and we should have a qualitative understanding of how cosmological perturbations may evolve. We now need to complete this analysis by defining the initial conditions, i.e. the seeds of structure, then characterizing how perturbations of different length scales evolve and finally identifying how we should ultimately characterise large scale structure today.

Over the decades there have been a plethora of proposals for the initial conditions of structure formation. One set of possibilities is that the Universe started off in a quasi-chaotic initial state and that the thermal initial state smoothed out the large inhomogeneities leaving a residue of fluctuations which then evolved to form structure. Clearly this is not a viable proposal unless we severely modify the nature of the Universe at early time – as we saw in the

previous section, structure on very large scales (larger than the Jeans wavelength) will tend to grow under the force of gravity. Furthermore, there is a causal limit to how much the Universe could homogenise, so it is in fact physically impossible to implement such a simple idea.

There is, however, a proposal that tends to smooth out the Universe and that changes the causal structure of space-time. Inflation [18–22] will take a microscopic patch of the Universe which is in thermal equilibrium and is well within the Jeans wavelength at that time, and expand it to macroscopic, cosmological proportions. In doing so, inflation solves the problem of how to homogenise the Universe on large scales but also provides a mechanism for seeding structure. We expect that, due to the quantum nature of space-time and matter, that the Universe will be riven by quantum fluctuations on microscopic scales. A period of inflationary expansion will amplify and stretch these quantum fluctuations to macroscopic scales which will be many times larger than the cosmological horizon by the time inflation ends. As the Universe resumes its normal expansion in the radiation era, the fluctuations will seed structure in the cosmological fluid which will then evolve in the manner described in the previous sections.

The form of the initial conditions arising from inflation have a deeply appealing feature: they will correspond to a *random field* which has an almost *scale invariant* gravitational potential. In this context, a random field is a three-dimensional function which can be generated through some random process; this should not come as a surprise given that the source of the fluctuations is a quantum process. And if you think about what we are trying to do, and look at the structure of the sky, you will realise that there must be an element of randomness. Our theory won't tell us if a cluster of galaxies, or a filament of galaxies or more generally an overdensity or underdensity is going to be at an exact position in space. All we can talk about is how much more probable structures of a given size are going to be relative to others. For example, we may expect to see more structure of 1 Mpc than of 100 Mpc, but we don't know exactly where they will be. Hence we talk about our density contrast, $\delta$, or gravitational potentials being random fields for which we can calculate their statistical properties.

We characterise a random field in much the same way we would characterise any other random process. For example we will have that the density contrast, $\delta$ satisfies

$$\langle \delta(\vec{x}) \rangle = 0,$$

where $\langle \cdots \rangle$ is an ensemble average, i.e. an average over all possible configurations of $\delta$. We can characterise its variance in terms of a *correlation function*, $\xi(r)$ through

$$\xi(|\vec{x} - \vec{x}'|) \equiv \langle \delta(\vec{x}) \delta(\vec{x}') \rangle,$$

or alternatively in terms of its *power spectrum*

$$\langle \delta_{\vec{k}} \delta_{\vec{k}'} \rangle = (2\pi)^3 P(k) \delta^{(3)}(\vec{k} + \vec{k}').$$

By defining $\xi(r)$ or $P(k)$ we can characterise the statistical properties of the random field[3]. It is often useful to consider the dimensionless version of the power spectrum, the *mass variance* which is given by

$$\Delta^2(k) = \frac{k^3 P(k)}{4\pi}.$$

We now need to understand what is meant by scale invariance. Let us define the average gravitational potential in a ball of radius $R$ to be

$$\Phi(R) = \frac{1}{V_R} \int_{V_R} d^3x \; \Phi(\vec{x}),$$

---

[3]This is only strictly true of the random process is Gaussian. For non-Gaussian processes one has to go further and characterise such quantities as $\langle \delta(\vec{x}_1) \delta(\vec{x}_2) \delta(\vec{x}_3) \rangle$ and higher order products. It turns out the inflation predicts that the random fields are, to a very good approximation, Gaussian.

where $V_R$ is the volume of the ball. We can define the variance of $\Phi$ on that scale to simply be

$$\sigma_R^2(\Phi) = \langle \Phi^2(R) \rangle.$$

A scale invariant spectrum corresponds to a variance which is independent of $R$, i.e. $\sigma_R^2(\Phi) \propto$ constant. It turns out that we can relate $\sigma_R^2(\Phi)$, to $\delta(t, \vec{k})$ through the Newton-Poisson equation. Indeed we have that

$$\sigma_R^2(\Phi) \simeq \frac{k^3}{4\pi} \langle |\Phi(t, \vec{k})|^2 \rangle \propto \frac{k^3}{k^4} \langle |\delta(t, \vec{k})|^2 \rangle \ \ \text{with} \ \ k = \frac{2\pi}{R}.$$

If it is scale invariant we then have that the power spectrum of the density fluctuations at initial time $t_i$, $P_i(\vec{k}) \equiv \langle |\delta(t_i, \vec{k})|^2 \rangle$ has the form $P_i(k) \propto k$. We can generalise the initial condition to include some scale dependence with $P_i(k) \propto k^{n_S}$, where we define $n_S$ to be the scalar spectral index. In practice, for calculational purposes, choosing initial conditions for the density field corresponds to picking the amplitude of the density field[4] to be given by $|\delta(t, \vec{k})| = \sqrt{A_S} k^{n_S/2}$.

## 10 Predicting the linear power spectrum of the matter distribution

Having chosen a set of initial conditions, we can now predict what the large scale structure of the Universe (approximately) looks like for different sets of assumptions. So far we have only studied the linear evolution of perturbations which is accurate on large scales and early times, before the non-linear regime ensues. Let us consider, for now, a universe with just radiation and cold dark matter. And let us approximate the evolution of cold dark matter in the RD era to be constant (and not $\propto \ln a$). We then have that for $k/\mathcal{H} \sim k\tau \gg 1$ that $\Delta_M \sim$ constant in the RD era and $\Delta_M \sim \tau^2$ in the MD era. If we follow a mode, $k$, at early times, so that $k\tau \ll 1$, it starts off outside the horizon. If $k > k_{\text{eq}}$, i.e. it is inside the horizon at equality, which means it will have entered the horizon during the RD era at a time $\tau_k = 1/k$. Such a mode has three stages of evolution. It grows from the initial time until $\tau_k$, then it is constant between $\tau_k$ until $\tau_{\text{eq}}$ and then it grows again until today, $\tau_0$. Putting it all together, we have that

$$\delta_M = \sqrt{A_S} k^{n_S/2} \left( \frac{\tau_0}{\tau_i} \right)^2 \frac{1}{(k\tau_{\text{eq}})^2}.$$

Modes that have $k < k_{\text{eq}}$ don't enter the horizon during the RD era and satisfy

$$\delta_M = \sqrt{A_S} k^{n_S/2} \left( \frac{\tau_0}{\tau_i} \right)^2.$$

Putting it all together we find that the power spectrum can be approximated by

$$P(k) = A_S \left( \frac{\tau_0}{\tau_i} \right)^2 \begin{cases} k^{n_S} & \text{if } k < k_{\text{eq}} \\ k_{\text{eq}}^4 k^{n_S-4} & \text{if } k \geq k_{\text{eq}} \end{cases} . \tag{3}$$

We can see that the resulting power spectrum has a different $k$ dependence on large and small scales. We can improve the accuracy of our prediction by including the fact that perturbations grow logarithmically in the RD era *and* that there is a contribution from baryons (which oscillated in the RD era). The result is a faint imprint of the oscillations on the $k > k_{\text{eq}}$ branch of the power spectrum as can be seen in Fig. 6. These are known as the Baryonic Acoustic Oscillations (BAOs). If we transform back to the correlation function, the effect of the baryons

---

[4]This is not strictly true, otherwise we would have $\langle \delta \rangle \neq 0$ but given that we are not interested in $\delta_{\vec{k}}$ today, but in $P(k)$, this prescription will suit us.

can be seen as a bump on scales of $r \simeq 105h^{-1}$ Mpc. Including a $\Lambda$ era won't change the qualitative features of the power spectrum (it will just affect the overall normalisation). The resulting power spectrum, with the appropriate choice of parameters, corresponds to the $\Lambda$ Cold Dark Matter power spectrum.

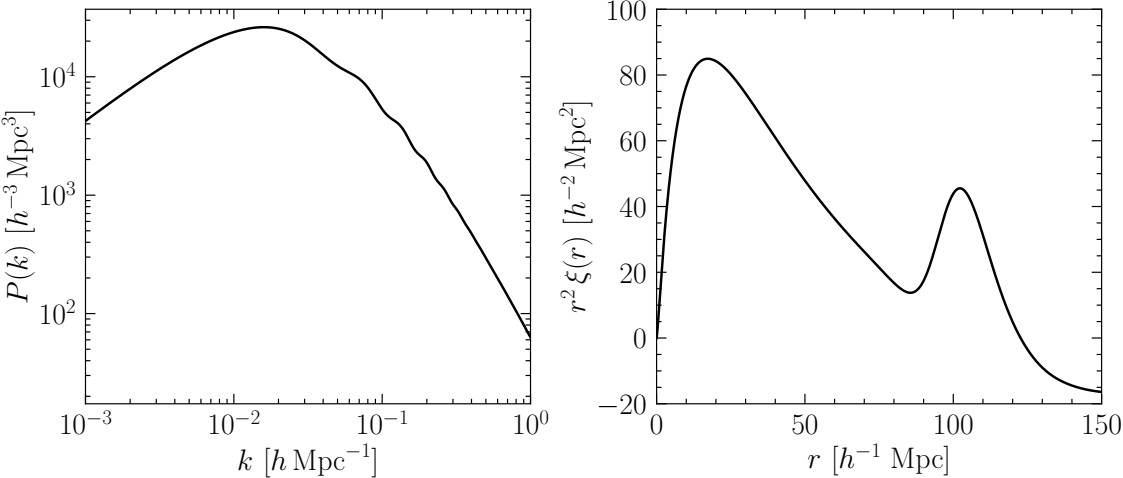

Figure 6: The linear matter power spectrum (left) and the corresponding correlation function (right) at $z = 0$ for $\Lambda$CDM.

It is interesting to consider a few other examples which, while ruled out by the data, can give us an idea how our choice of physical model will affect the power spectrum. The simplest scenario we can imagine is a flat Universe consisting solely of photons, atoms (i.e. Baryons) and a cosmological constant. There is a limit on how much of the Universe can be made of baryons: the abundance of light elements restricts $\Omega_b h^2 \simeq 0.024$. With our current constraints on the Hubble constant, this means that fractional energy density in baryons must be around 5% and given that we are considering a Universe with $\Omega = 1$ we must have $\Omega_\Lambda = 0.95$. We call this cosmology $\Lambda$BDM and in Fig. 7 we plot the power spectrum of such a theory. We can clearly identify the main features. On very large scales (i.e. on scales larger than the sound horizon at equality between matter and radiation), perturbations will grow until they reach the $\Lambda$ era, after which they will be constant. On scales below the sound horizon, perturbations will initially grow, then oscillate acoustically and once the Universe recombines, they will grow again until the freeze in during the $\Lambda$ era. Hence we see a series of peaks and troughs on intermediate to small scales. On very small scales, i.e. on scales which are smaller than the Silk damping scale at recombination, perturbations are severely suppressed and we can see exponential damping. Another simple model we can consider is one where we replace the pressureless matter in the CDM model by light massive neutrinos. The motivation is clear: we know that neutrinos exist and there is even evidence that they may have a mass. As we saw in the previous section, neutrinos will not evolve as a fluid and will free stream while they are relativistic, exponentially damping all perturbations on small scales. The neutrinos are weakly interacting, dark (i.e. they don't interact strongly with light) and move relativistically so can be considered a 'hot' component of the Universe. For these reasons, a Universe in which neutrinos make up the bulk of the dark matter today is called the $\Lambda$ Hot Dark Matter scenario or $\Lambda$HDM. We plot the power spectrum for this theory in Fig. 7. Clearly, these different models have very distinct predictions depending on the parameters that we use. Which means that, if we can measure the power spectrum, we can rule in or out different models of large scale structure.

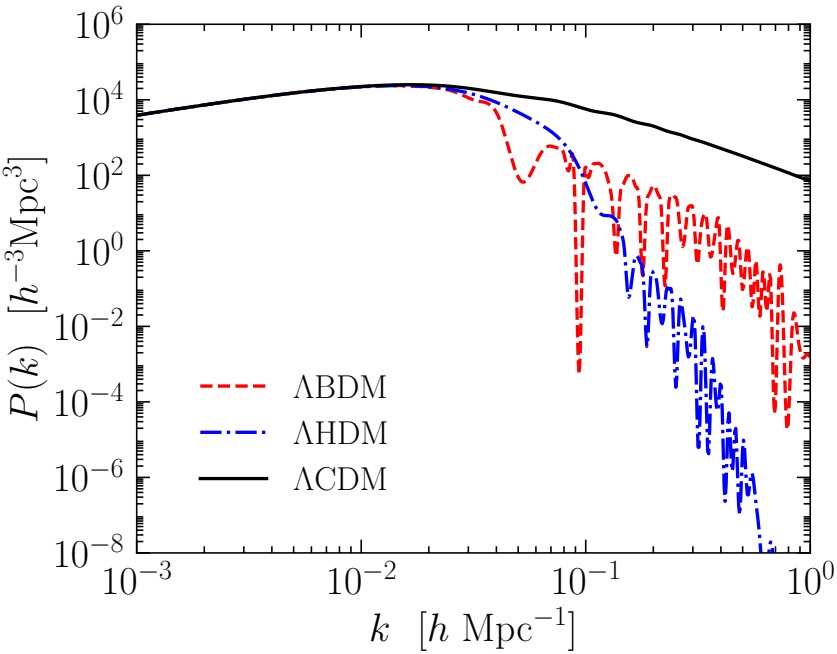

Figure 7: Models of large scale structure with different types of dark matter: $\Lambda CDM$ has cold dark matter, $\Lambda BDM$ has baryonic dark matter and $\Lambda HDM$ has massive neutrinos.

## 11 Non-linear evolution of large scale structure

We have used linear perturbation theory to make predictions, but we know that gravity is non-linear. This particularly obvious if we look at the conservation of energy and momentum equations. If we separate these equations in a linear and a non-linear part, we have that

$$
\begin{aligned}
\dot{\delta} + \nabla \cdot \vec{v} &= -\vec{\nabla} \cdot (\delta \vec{v}), \\
\frac{d}{dt}(\vec{\nabla} \cdot \vec{v}) + 2H\vec{\nabla} \cdot \vec{v} + \frac{3}{2}H^2 \delta &= -\vec{\nabla} \cdot \left[ \vec{v} \cdot \vec{\nabla} \vec{v} \right].
\end{aligned}
$$

One way to solve these equations is to use perturbation theory again, this time including higher order terms so that

$$
\begin{aligned}
\delta &= \delta_1 + \delta_2 + \cdots, \\
\vec{v} &= \vec{v}_1 + \vec{v}_2 + \cdots,
\end{aligned}
$$

and then solve order by order.[5]  So for example, the conservation of mass equation becomes hierarchy of equations:

$$
\begin{aligned}
\dot{\delta}_1 + \nabla \cdot \vec{v}_1 &= 0, \\
\dot{\delta}_2 + \nabla \cdot \vec{v}_2 &= -\vec{\nabla} \cdot (\delta_1 \vec{v}_1), \\
&\cdots
\end{aligned}
$$

and the same again with the Euler equation. One can solve this hierarchy order by order and with it, it is possible to probe the mildly non-linear regime but one finds that it very rapidly breaks down and that it is impossible to go very far.

---

[5]Several approaches exist to do this including standard perturbation theory [23] and effective field theory [24].

To see how rapidly perturbation theory breaks down, we can embrace the full non-linearity and look at what happens to an isolated patch of the universe as it undergoes non-linear collapse. One can model it as an isolated, closed universe, with a density greater than the background (critical) density. The solutions to such a patch are well-known. If we look at a patch of size $R$ with mass $M$, a particle of unit mass with total energy $E$ obeys

$$\frac{\dot{R}^2}{2} - \frac{GM}{R} = E,$$

and we can define the density to be $\rho = 3M/r\pi R(t)^3$. Now, we are interested in comparing the evolution with the background density (in a matter dominated) universe. We have that the background density at some initial time is given by $\bar{\rho}_i = 3M/4\pi R_i^3$ and, as a function of time, $\bar{\rho} \propto 1/a^3 \sim (t_i/t)^2$.

There is a parametric solution for $R$ and $t$ in terms of $\vartheta$

$$\begin{aligned} R &\sim& 1 - \cos\vartheta, \\ t &\sim& \vartheta - \sin\vartheta, \end{aligned}$$

which, when all put together leads to a solution for the linear and non-linear density contrasts:

$$\begin{aligned} \delta_L &=& \frac{3}{5}\left[\frac{3}{4}(\vartheta - \sin\vartheta)\right]^{2/3}, \\ \delta_{NL} &=& \frac{9}{2}\frac{(\vartheta - \sin\vartheta)^2}{(1 - \cos\vartheta)^3} - 1. \end{aligned}$$

Comparing the two, we find that $\delta_{NL}$ diverges when $\delta_L \simeq 1.686$. In other words, perturbation theory breaks down very rapidly. If we include the fact that matter has some velocity, we find that a virialised sphere (where its potential energy, $V$ and kinetic energy, $K$ satisfy, $V = -2K$), at the time of collapse, will have a density 178 times larger than the background density.

In practice, the only way to properly explore the non-linear regime is to simulate it [25]. This means solving the evolution of a collection of point masses that represent chunks of the matter fluid, under the influence of their gravitational interactions. Specifically, one needs to solve

$$\ddot{\vec{x}}_i + 2H\dot{\vec{x}} = -\nabla\Phi_i$$

and either the Newton-Poisson equation

$$\nabla^2\Phi = 4\pi G\rho,$$

where $\rho$ is some smoothed, binned density field constructed from the point particles, or a direct expression for $\Phi$,

$$\Phi(\vec{x}) = -G\sum_i \frac{m_i}{|\vec{x} - \vec{x}_i|},$$

or a combination of the two. Such an approach is limited by the box size (i.e. how large a scale can be accurately modelled) and the number of particles (affecting the precision on small scales). It does, however, allow for the inclusion of baryonic and astrophysical processes which are essential to model the formation and evolution of galaxies and gas. Some of the most popular codes and simulation suites include IllustrisTNG [26], Abacus [27], Quijote [28] and CAMELS [29]. We note, however, that there is still a large uncertainty in the accuracy with which one is able to model these non-gravitational processes. While different codes may lead to the same qualitative features, they can disagree between each other by 100%.

Finally, one way of modelling nonlinear structure formation, incorporating information from N-body simulations is the *halo model* [30]. In this approach, all matter is assumed to reside in gravitationally bound dark matter *halos* of various masses[6]. Each halo is a virialised object containing a certain mass of dark matter, and the entire matter distribution can be built up by considering the population of halos and their properties. The halo mass function, $n(M)$, gives the comoving number density of halos of mass $M$. The mass function is often expressed in terms of the dimensionless peak height $\nu$, defined by $\nu \equiv [\delta_c/\sigma(M)]^2$, where $\delta_c \approx 1.686$ (the number we found before in spherical collapse) and $\sigma(M)$ is the rms linear density fluctuation on the scale corresponding to mass $M$. In the Press–Schechter (PS) formulation, the fraction of mass in halos of mass $M$ is often written as $\nu f(\nu)$, with:

$$\nu f(\nu) = \sqrt{\frac{\nu}{2\pi}} \exp\left(-\frac{\nu}{2}\right),$$

which leads to the PS mass function. In terms of $n(M)$, this implies

$$\frac{dn}{dM} = \frac{\bar{\rho}}{M} f[\nu(M)] \frac{d\nu}{dM}.$$

The halo density profile $\rho(r|M)$, describes the internal mass distribution within a halo of mass $M$. For simplicity, halos are usually assumed to be spherically symmetric, with $\rho(r|M)$ decreasing with radius. A common parametrisation is the Navarro–Frenk–White (NFW) profile [33], measured in N-body simulations,

$$\rho_{\mathrm{NFW}}(r|M) = \frac{\rho_s}{(r/r_s)\left(1 + r/r_s\right)^2},$$

where $r_s$ is a characteristic inner radius (scale radius) and $\rho_s$ is a corresponding density scale. Each halo is truncated at a virial radius $R_{\mathrm{vir}}$, beyond which $\rho(r)$ is effectively zero (this ensures the halo has finite mass).

It is often useful to define the *concentration* $c(M) \equiv R_{\mathrm{vir}}/r_s$, which typically depends on halo mass and redshift: lower-mass halos tend to be more concentrated (larger $c$) than massive halos formed more recently. Halos are biased tracers of the overall matter distribution. Massive halos tend to cluster more strongly than the dark matter as a whole. The halo bias $b(M)$ is defined such that, on large linear scales, the overdensity of halos of mass $M$ is $b(M)$ times the overdensity of matter: $\delta_h(M) \approx b(M)\delta_m$ (for $\delta_m \ll 1$).

Using the above ingredients, the halo model provides an expression for the matter power spectrum $P(k)$ as a sum of two contributions

$$P(k) = P_{1h}(k) + P_{2h}(k),$$

where

$$P_{1h}(k) = \frac{1}{\bar{\rho}^2} \int_0^\infty dM\, n(M)\, M^2\, |u(k|M)|^2,$$

and

$$P_{2h}(k) \simeq \left[\frac{1}{\bar{\rho}} \int_0^\infty dM\, n(M)\, M\, b(M)\, u(k|M)\right]^2 P_{\mathrm{lin}}(k),$$

where $u(k|M)$ is the Fourier transform of the halo density profile and we have factorised the two halo power spectrum, $P_{hh}(k|M_1, M_2) \approx b(M_1)\, b(M_2)\, P_{\mathrm{lin}}(k)$, where $P_{\mathrm{lin}}(k)$ is the linear power spectrum we derived in the previous section. We have that $P_{2h}(k \to 0) \to P_{\mathrm{lin}}(k)$,

---

[6]Halos are identified in simulations with algorithms such as the Friends-of-Friends algorithm [31] or the more modern ROCKSTAR algorithm [32].

meaning the halo model correctly reproduces the linear power spectrum on large scales as expected. The scale where $P_{1h}$ and $P_{2h}$ are comparable is often around the quasi-linear regime ($k \sim 1$–$10\, h/$Mpc), and is sometimes referred to as the "1-halo to 2-halo transition" scale. The halo model predictions for $P(k)$ are qualitatively and even quantitatively successful across a range of scales, but they are not exact. In practice, simple halo model calculations show small deviations from the precise results of cosmological $N$-body simulations, especially near the transition regime and at very high $k$.

The end result of non-linear growth is to boost the amplitude of fluctuations on small scales and change the shape of the power spectrum. In Fig.8, we superpose the linear and non-linear power spectra for $\Lambda$CDM.

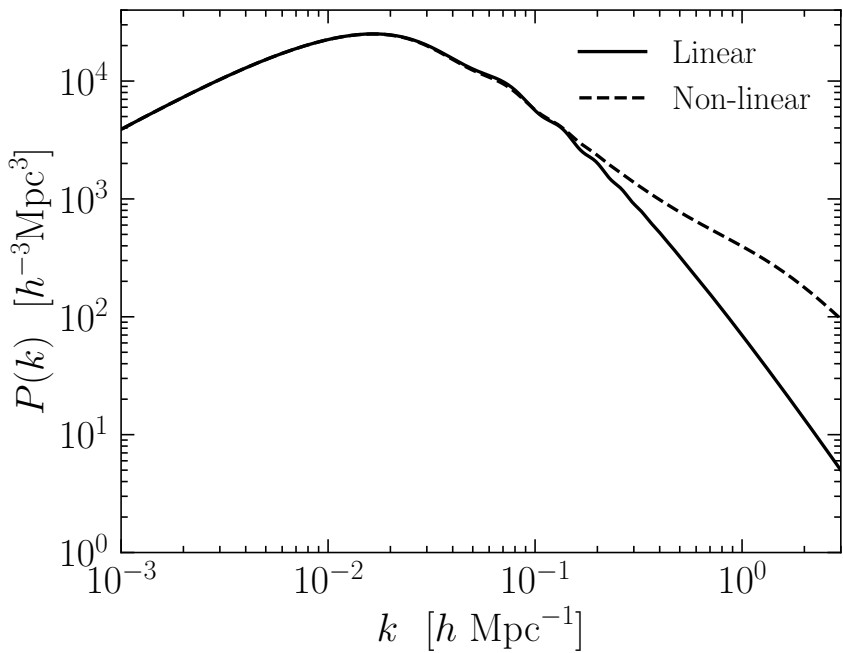

Figure 8: The linear and non-linear matter power spectra for $\Lambda$CDM at $z = 0$.

## 12 Measuring and constraining the expansion of the Universe

We now have the machinery to create model Universes and we would like to pin down which set of cosmological parameters (like $H_0$, $\Omega_M$, $\cdots$) correspond to *our* Universe. We *observe* the Universe by (mostly) collecting photons which have a certain frequency, or frequency range, with a certain intensity or energy. This tells us information about what emitted them and the environment they traveled through.

Let us first focus on the background evolution and try to constrain it. To do so, we need to measure distances and redshifts accurately. By far the easiest quantity to measure is the redshift. By looking at the shift in the spectra of known elements, it is possible to infer the recession velocity of the galaxy directly. We can see this in Fig. 9, where schematically show how a galaxy spectrum will be redshifted.

Measuring distances is much harder and we will now work through a few methods. The most direct method is to use parallax to measure the distance to a star. Let us remember what you do here. Imagine that you look at an object in the sky, at a distance $D$. It can be described in terms of two angles which define its position on the celestial sphere. Now imagine that we move a distance $2d$ from where we were. The object will be displaced an angle $\theta$ from

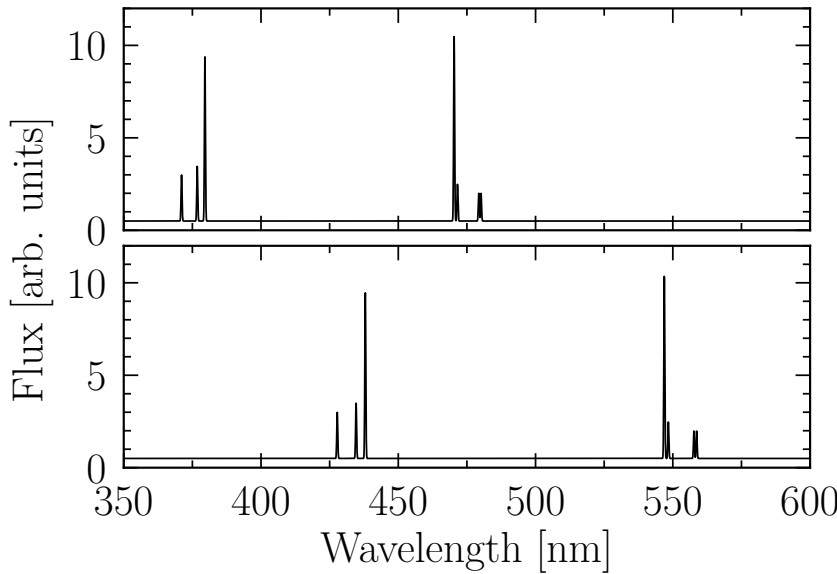

Figure 9: Reference spectral lines determined in a laboratory (top) compared to the same spectral lines observed at a redshift of $z = 0.2$ (bottom).

where it was. The angle that it has been displaced by will be related to the distance $D$ and displacement $d$. If we say $\theta = 2\alpha$, then we have $\tan \alpha = \frac{d}{D}$. If $\alpha$ is small, then we can use the small angle approximation to get $\alpha = d/D$.

The motion of the earth around the sun gives us a very good baseline with which to measure distance. The distance from the earth to the sun is 1 AU, so we have that $D = \frac{1}{\alpha}$, where $\alpha$ is measured in arcseconds. $D$ is then given in *parsecs*. One parsec corresponds to $206,265$ AU or $3.09 \times 10^{13}$ km. This is a tremendous distance, 1 pc $\sim 3.26$ light years. All stars have parallax angles less than one arcsecond. The closest star, Proxima Centauri, has a distance of 1.3 pc. The Gaia satellite [34], launched by the European Space Agency, has measured the positions, motions, and brightnesses of over 1.8 billion stars in our galaxy with unprecedented precision. Using parallax, Gaia can directly measure distances to stars up to about $10,000$ light-years, or roughly 3 kiloparsecs, with good accuracy for most stars. For the brightest and nearest stars, parallax measurements are precise even out to 10 kiloparsecs or more, though uncertainties increase with distance. Gaia's data has dramatically refined our three-dimensional map of the Milky Way.

We would like to be able to look further. The basic tool for doing this is to take an object of known brightness and see how bright it actually looks. For example, take a star with a given luminosity $L$. The luminosity is the amount of light it pumps out per second. How bright will it look from where we stand? We can think of standing on a point of a sphere of radius $D$ centred on the star. The brightness will be $B = \frac{L}{4\pi D^2}$. The further away it is, the dimmer it will look. If we know the luminosity of a star and we measure its brightness, then we will know how far away it is.

How can we use the luminosity distance to move out beyond the 10 kpc we can reach with parallax? There are some stars which have a very useful property. Their brightness varies with time and the longer their variation, the larger their luminosity. These stars known as *Cepheid* stars are interesting because they have a) periods of days (which means their variations can be easily observed) and b) are very luminous with luminosities of about $100 - 1000 L_\odot$ (where $L_\odot$ is the luminosity of the Sun) and therefore they can be seen at great distances. It was found that their period of oscillation is directly related to their intrinsic luminosity. These stars pulsate because their surface oscillates up and down like a spring. The gas of the star

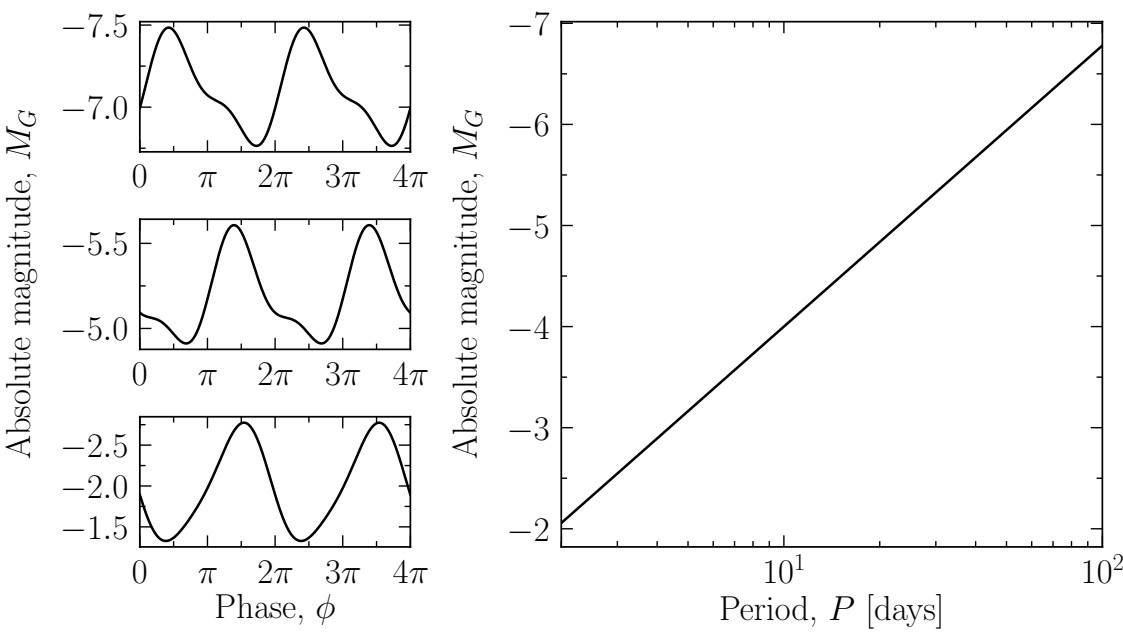

Figure 10: The luminosity of Cepheid stars varies periodically over time (left). There is a tight relationship between the period (x-axis) and the luminosity (or magnitude in the y-axis) for Cepheid and RRLyrae star (right), given by $M_G = -2.78 \log_{10}(P) - 1.22$.

heats up and then cools down, and the interplay of pressure and gravity keeps it pulsating. How do we know the intrinsic luminosity of these stars? We pick out globular clusters (very bright aglommerations in the Galaxy with about $10^6$ stars) and we use parallax to measure their distances. Then we look for the varying stars, measure their brightness and period and build up a plot. This is shown in Fig. 10.

The method of choice for measuring very large distances is to look for distant *supernovae*. Supernovae are the end point of stellar evolution, massive explosions that pump out an incredible amount of energy. Indeed supernovae can be as luminous as the galaxies which host them with luminosities of around $10^9 L_\odot$. A certain type of supernova (supernovae $I_A$) seem to have very similar behaviours. Supernovae $I_A$ arise when a white dwarf which is just marginally heavier than the Chandrasekhar mass gobbles up enough material to become unstable and collapse. The electron degeneracy pressure is unable to hold it up and it collapses in a fiery explosion. They are extremely rare, one per galaxy per hundred years, so we have to be lucky to find them. However, there are $10^9$ galaxies to look at, so the current practice is to stare at large concentrations of galaxies and wait for an event to erupt. So we can see distant supernovae, measure their brightness and if we know their luminosities, use the inverse square law to measure the distance. In practice they don't all have the same luminosities, but the rate at which they fade after explosion is intimately tied to the luminosity at the moment of the explosion. Specifically, the Phillips relation [35] gives a correlation between the peak amplitude and the amount by which the luminosity has decayed after 5 days. So by following the ramp up to the explosion and the subsequent decay it is possible to recalibrate a supernova explosion so that we know its luminosity to within 5%. This is shown in Fig. 11.

Supernovae can be used to measure distances out to large distances, which make them a very powerful probe of the background expansion as one can break away from the *peculiar velocities* due to the local gravitational field arising from overdensities and underdensities. In particular, we have that Hubble's law will be contaminated $z = H_0 d + v_{pec}$, where the peculiar velocities can take values of $v_{pec} \sim 300 - 600$ km s$^{-1}$, obscuring any attempts at measuring

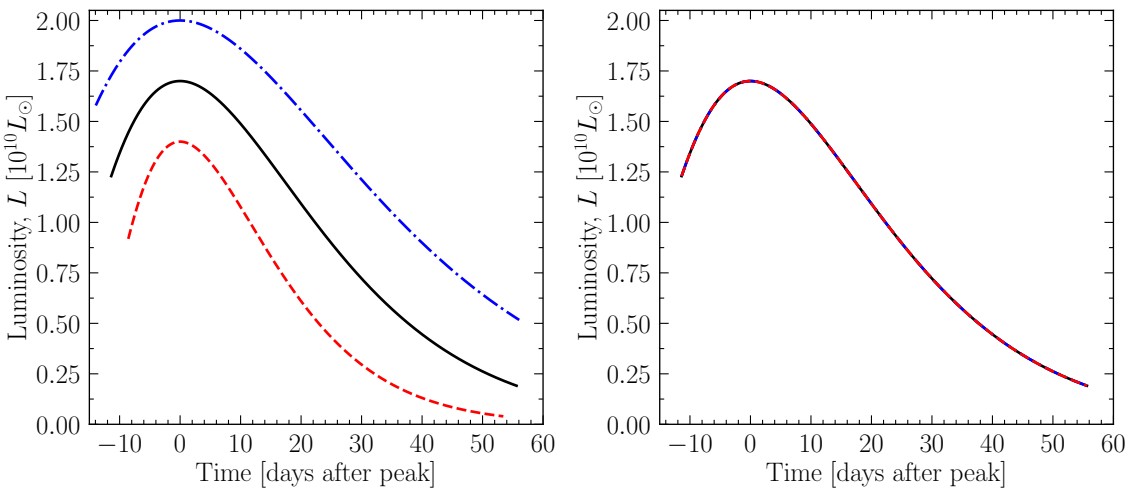

Figure 11: Supernovae $I_A$ luminosity curves as a function of the number of days after the peak intensity (left) and the same supernovae which lie one on top of another after correcting for colour and stretch, enabling their use as a standard candle (right).

$H_0$.

The use of supernovae to constrain the expansion of the Universe has been immensely successful, primarily led by the 'SHOES' collaboration [36] which looks at various stages of the distance ladder, including cepheids and SN $I_A$. But there are major concerns (apart from the fact that they are not true standard candles). These are: a) uncertainties in photometry (i.e. making sure you have correct brightnesses for each object); b) crowding and dust extinction (other sources affecting the brightness or dust reducing the brightness); c) the effect of metallicity in Cepheids, changing the period luminosity relation; d) the dependence of SN $I_A$ on the evolution of the host galaxies.

Another approach is to look at the the tip of the red giant branch (TRGB). In the Hertzsprung-Russell diagram, stars follow reasonably well-defined tracks on the luminosity-temperature plane. In particular old, low metallicity stars have very consistent tracks in the infra-red. You can observe them in halos and find a sharp drop at approximately $-4.05$ magnitudes in the I-band (see Fig. 12). The idea is to then use these objects, instead of Cepheids, to anchor distant supernovae. The Carnegie-Chicago-Hubble Program (CCHP) has pioneered this method to great effect [37, 38]

We can turbo-charge this exercise and try to go beyond the linear relationship between distance and redshift. As we saw, to quadratic order we have $cz \simeq H_0 d - q_0 (H_0 d)^2/2$, where $q_0 = \Omega_M/2 - \Omega_\Lambda$. To go beyond the local universe we need to go to high redshifts. We can do that with SN $I_A$ and that was done, to great effect, in the late 1990s, leading to the first direct evidence for accelerated expansion [39, 40], i.e. $q_0 < 0$.

Another method for constraining the expansion of the Universe is by using the remnant of the baryons in the power spectrum of density perturbations. We will look at such observables more carefully later on, but we can address their use for background cosmology here. As mentioned in our discussion of the linear power spectrum, the scale of the acoustic oscillations in the baryons is set by acoustic horizon at decoupling (when the baryons decouple from the photons, close to recombination). This is a fixed, comoving scale, $r_d$, which is then imprinted in the power spectrum of the matter perturbations. As we saw, it is sub-dominant, accounting for small oscillations which are of order a few percent around the overall shape of the power spectrum.

If we now measure the power-spectrum, at a given redshift, such a comoving scale will

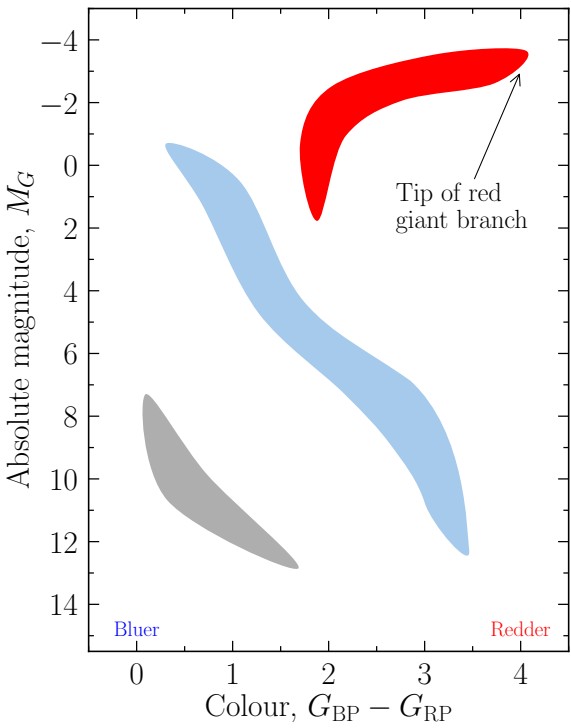

Figure 12: Schematic Hertzsprung-Russell diagram showing the tip of the red giant branch.

subtend a certain angular size (that we can measure). And if we have a fixed comoving scale and an angle, we can infer, as we saw, a comoving angular diamater distance, $\chi$. Alternatively, if we measure such a distance along the line of site, we have the scale of the oscillation in redshift, $\Delta z$ is related to the local comoving scale via $c\Delta z \simeq H(z)r_d$. In other words, we have a local measurement of the Hubble expansion rate which, as we know, is related to the cosmological parameters. With these two different ways of measuring $H(z)$ it is possible to reconstruct the time dependence of the expansion rate of the Universe and, as a result, infer the cosmological parameters.

## 13    Surveys and observables

So far we have focused on the background evolution and attempting to measure the expansion rate of the Universe, extending the program that Edwin Hubble initated a century ago. But modern cosmology aspires to do more and map out the large scale structure of the Universe. To do so involves looking at the sky (celestial sphere), in one way or another. If we observe the sky, we end up with a map $M$ which is a function of the direction in the sky, $\hat{n}$, $M(\hat{n})$. It is useful to expand this map in terms of the spherical harmonics $Y_{\ell m}(\hat{n})$ so that

$$M(\hat{n}) = \sum_{\ell m} a_{\ell m} Y_{\ell m}(\hat{n}).$$

The $a_{\ell m}$ are the 'Fourier coefficients' of the map on the sphere. As discussed above, in cosmology we often assume that the cosmological signal of interest is stochastic and that this is an 'ensemble', such that we are observing a 'draw' or realisation of the ensemble. The maps we look at are fluctuations so

$$\langle M(\hat{n}) \rangle = 0.$$

We also promote homogeneity and isotropy to *statistical* homogeneity and isotropy. This means that

$$\langle M(\hat{n})M(\hat{n}')\rangle = C(\hat{n}\cdot\hat{n}').$$

It does not depend on the direction or orientation.

Finally, it is often assumed and can be directly motivated, that on the largest scales these fluctuations are Gaussian. This means that they are described by a multivariate Gaussian distribution. To understand what this means, let us assume that we discretise our sky into a number of discrete cells or "pixels" $\hat{n}_p$, where $p \in 1,\dots,N$, and we define $M(n_p)\equiv M_p$. Then we can define the covariance matrix

$$\langle M_p M_{p'}\rangle = \mathbb{C}_{pp'}.$$

We can construct the multivariate Gaussian

$$\mathcal{P}[M||\mathbb{C}] = \frac{(2\pi)^{N/2}}{\det^{1/2}\mathbb{C}}\exp\left(-\frac{1}{2}M^T\mathbb{C}^{-1}M\right),$$

where we have collected the data into a pixel data vector $M = (M_1,\dots,M_N)$.

Let us go back to the Fourier representation. We now have

$$\langle a_{\ell m}\rangle = 0.$$

Statistical isotropy gives us

$$\langle a_{\ell m}a^*_{\ell'm'}\rangle = C_\ell\ \delta_{\ell'\ell}\delta_{m'm},$$

where $C_\ell$ is the *angular power spectrum*. An interesting thing to ask is how well can we measure $C_\ell$? If you think that you can measure a finite number of modes for each $\ell$ ($2\ell + 1$ values of $m$), you can see that the lowest $\ell$ have fewer $m$ modes than the largest $\ell$.

A simple "estimator" of the angular power spectrum is to consider

$$\hat{C}_\ell = \frac{1}{2\ell+1}\sum_m |a_{\ell m}|^2.$$

If we have a full sky map, this estimator is unbiased in the sense that

$$\langle \hat{C}_\ell\rangle = C_\ell.$$

The variance is

$$\sigma^2(\hat{C}_\ell) = \frac{2C_\ell^2}{2\ell+1}.$$

We observe that the variance is highest for low $\ell$. This is known as *Cosmic Variance*.

Our Universe is three dimensional so what are we actually seeing, when we look at a map on the sphere, is the integral of a three-dimensional field along the line of sight. For example, if the three dimensional field is $\Delta(\vec{x},t)$ we have that

$$M(\hat{n}) = \int \mathrm{d}\chi\ q(\chi)\Delta(\chi\hat{n},t(\chi)),$$

where $\chi$ is the comoving distance, $q(\chi)$ is the radial kernel (which depends on how we are actually observing the three dimensional field) and $\Delta(\vec{x},t)$ is measured along the line of sight. If we Fourier transform $\Delta(\vec{x},t)$, so $\Delta(\vec{x},t)\to\tilde{\Delta}(\vec{k},t)$, we can define the power spectrum

$$\langle\tilde{\Delta}(\vec{k},t)\tilde{\Delta}(\vec{k}',t)\rangle = (2\pi)^3 P_\Delta(k)\delta^{(3)}(\vec{k}+\vec{k}').$$

One can show that, with the Limber approximation [41][7],

$$C_\ell \simeq \int \frac{\mathrm{d}\chi}{\chi^2} (q(\chi))^2 P_\Delta \left( k = \frac{\ell + 1/2}{\chi}, z(\chi) \right).$$

Each three-dimensional scale $k$ is mapped onto a two-dimensional scale $\ell$.

Another important ingredient, when constructing a map of the large scale structure of the Universe, is the motion of photons from the source to the observer (the telescope) in an inhomogeneous Universe. We have already used the fact that the photon redshifts in a smooth, expanding background. To go beyond that, we need to look, in more detail, at light rays in a perturbed Universe. Consider a photon geodesic, $x^\mu(\lambda)$, where $\lambda$ is the affine parameter of the geodesic. The photon has momentum

$$P^\mu = \frac{\mathrm{d}x^\mu}{\mathrm{d}\lambda}.$$

The geodesic equation is

$$\dot{P}^\mu + \Gamma^\mu{}_{\alpha\beta} P^\alpha P^\beta = 0$$

and it is light-like so

$$P^\mu P^\nu g_{\mu\nu} = 0.$$

Consider the perturbed metric we defined above:

$$ds^2 = a^2(\tau) \left[ -(1 + 2\Phi) dt^2 + (1 - 2\Phi) d\vec{x}^2 \right].$$

And consider a comoving observer on this perturbed space time. It has a time-like vector

$$U^\mu = \frac{1}{a} (1 - \Phi) \delta_0{}^\mu.$$

We can define two useful quantities $\epsilon \equiv a P_\mu U^\mu$ and $\hat{e} \equiv \vec{P}/|P|$. Both of these quantities are conserved on a homogeneous background. We can rewrite the components of $P^\mu$ as

$$
\begin{aligned}
P^0 &= \frac{1}{a^2} \epsilon (1 - \Phi), \\
\vec{P} &= P_0 (1 + 2\Phi) \hat{e},
\end{aligned}
$$

to find

$$
\begin{aligned}
\frac{1}{\epsilon} \frac{d\epsilon}{d\tau} &= -\frac{d\Phi}{d\tau} + 2\Phi', \\
\frac{d\hat{e}}{d\tau} &= -2\vec{\nabla}_\perp(\Phi),
\end{aligned}
$$

where $\vec{\nabla}_\perp = \vec{\nabla} - \hat{e}\left(\hat{e} \cdot \vec{\nabla}\right)$ is the transverse gradient, across the line of site.

We can integrate to find

$$
\begin{aligned}
\frac{\epsilon}{\epsilon_0} &= 1 + \Phi_O - \Phi + 2 \int_\tau^{\tau_0} d\tau' \Phi', \\
\vec{x}(\tau) &= -\hat{e}_O \int_\tau^{\tau_0} d\tau' (1 + 2\Phi) - 2 \int_\tau^{\tau_0} d\tau' (\tau - \tau') \vec{\nabla}_\perp \Phi.
\end{aligned}
$$

---

[7]Note: We can change $\mathrm{d}\chi \to \mathrm{d}z/H$. Also $\mathrm{d}\eta = \mathrm{d}\chi$.

As we have seen, redshifts play an important role and we measure them by comparing photon frequencies. We have that $h\nu = P^\mu U_\mu = \epsilon(1 + \hat{n} \cdot \vec{v})/a$, where the last terms captures the Doppler effect. Taking the ratio between two frequencies (now and in the past), we have

$$
\begin{aligned}
1 + z &= \frac{h\nu}{h\nu_0} = \frac{1}{a}\frac{\epsilon}{\epsilon_0}\frac{1 + \hat{n}\cdot\vec{v}}{1 + \hat{n}\cdot\vec{v}_0} \\
&= \frac{1}{a}\left[1 + \Phi_0 - \Phi + \hat{n}\cdot\vec{v} - \hat{n}\cdot\vec{v}_0 + 2\int_{\tau_0}^{\tau}d\tau'\Phi'\right].
\end{aligned}
$$

# 14  Galaxy surveys

One of the simplest things one can do is to think of galaxies as test particles and to count them [42]. The part of the galaxies you see directly, is made of baryons (in the form of stars or gas) and it should reside in a large concentration of dark matter, the halo. Where you have more galaxies, you expect there to be an over-density, i.e. a higher concentration of matter. Suppose, indeed that you count the number of galaxies per unit volume, and use it to define a local galaxy number density, $n_G(\vec{x}, t)$. If you assign each galaxy more or less the same mass you can then define a galaxy density contrast:

$$
\delta_G(\vec{x}, t) = \frac{n_G(\vec{x}, t) - \bar{n}_G(t)}{\bar{n}_G(t)},
$$

and use it as a proxy for the density contrast we studied when we were developing the theory.

If we think this through, we can rapidly see that it can't be that simple. A galaxy is formed through a somewhat elaborate process. Indeed a local overdensity will collapse to form a bound structure but other physical processes will come into play: gas will heat up and accrete, there will be torque from the local tidal field, chemical processes will be triggered, shocks will form as the gas swirls around all of which contribute to form, for example a spiral galaxy. So while we can say, schematically, that

$$
\delta_G = F[\delta_M],
$$

it is quite likely that the functional (not function) $F[\cdots]$ can be non-linear, possibly non-local and furthermore, stochastic [43]. The simplest assumption for this relationship is that it is linear on large scales

$$
\delta_G(\vec{x}, z) = b(z)\delta_M(\vec{x}, z)
$$

and we call $b(z)$ *linear bias*. A more general form is (where we assume all operation are local)

$$
\delta_G = b_1\delta_M + \frac{b_2}{2}\delta_M^2 + b_{S^2}S^2,
$$

where the tidal tensor is defined as

$$
S_{ij} = \left(\frac{\partial_i\partial_j}{\nabla^2} - \frac{1}{3}\delta_{ij}\right)\delta_M.
$$

From now on, we will restrict ourselves to linear bias.

Let us, in the light of what we said of what we actually observed in cosmology, think of what a galaxy survey looks like. We measure angles in the sky and we use redshifts, $z$ as a

proxy for radial distance. Then we can think of a two dimensional map of the galaxy density contrast as

$$\delta_G(\hat{n}) = \int dz p(z) \delta_G[\chi(z)],$$

where $p(z)$ is the radial distribution of galaxies, as a function of redshift. We can transform to $\chi$ to get

$$\delta_G(\hat{n}) = \int d\chi q_G(\chi) \delta_M(\chi\hat{n}, \chi),$$

where the window function (including the change of variables from $z \to \chi$) is

$$q_G(\chi) = b(z)H(z)p(z),$$

where $z$ depends on $\chi$.

*Spectroscopic* surveys pick well-known spectral lines, measure them and use them to determine the redshifts of each galaxy. These lead to incredibly precise redshifts with uncertainties, $\delta z < 10^{-3}$. As a result, we map a scale from three dimensions almost exactly onto a scale in two dimensions. Now recall that, one of the ways we can map out the expansion of the Universe is by looking at the Baryonic Acoustic Oscillations (the BAOs) – faint oscillations in the matter power spectrum. These are picked up by the galaxy power spectrum and, because spectroscopic surveys are so precise it is possible to pick up the oscillations in the angular power spectrum through $\Delta\ell_{\mathrm{BAO}} = 2\pi D_A/r_d$ where $r_d$ was defined above. It is also possible, because of the precise redshift measurement, to constrain the local Hubble rate $\Delta z_{\mathrm{BAO}} = r_d H(z)$. This is one of the main methods used to measure the expansion rate of the Universe as a function of time, as first discussed previously [44].

There is an added complication which is also a source of information. As we saw above, the local gravitational field will induce a local peculiar velocity, which will affect the redshift. So we will have that the observed redshift, $z^O$, will be the 'true' redshift, $z$, plus a correction, $z^O = z + v_p$, where $v_p = \hat{n} \cdot \vec{v}/a$. This, in turn, will lead to an error in the distance as $H_0 r^O = z + v_p = H_0 r + H_0 \delta r$, i.e. this correction will distort what we infer to be the local galaxy density contrast. In particular, the redshift space galaxy density contrast is related to the real space one (at linear order) via

$$\delta_G^s = \delta_G - \frac{1}{H}\partial_r v_p.$$

We can use the conservation of mass equation

$$\dot{\delta} + \vec{\nabla} \cdot \vec{v} = 0,$$

to determine an expression for galaxy density contrast in Fourier space

$$\delta_G^s(\vec{k}) = (b + \mu^2 f)\delta_M(\vec{k}),$$

where $\mu = \hat{k} \cdot \hat{n}$ and we have defined the *growth rate*, $f = d\ln\delta_M/d\ln a$. This correction to the density contrast on linear scales is known as *redshift space distortion* (or RSD for short) [45]. The effect on the correlation function of the galaxy density contrast is to squash it along the line of sight on large scales but to lead to spikes on small scales (known as the fingers of God effect). Interestingly, because it has such a distinct signature, we can pull it out of the data and use it to measure $f$ or, more usually, $f\sigma_8$ which, in turn, will also depend on cosmological

parameters. While the peculiar velocities are a contaminant, they are a useful contaminant which contain additional information.

We must now address the main imperfections when analyzing a spectroscopic redshift survey. An obvious one is what we call *Cosmic Variance*. On the largest scales we have fewer Fourier modes to sample the density field. To understand this limitation, one has to realise that we only see a finite patch of the Universe which in turns means that the Fourier modes are effectively discretised. That means we only measure the power spectrum at a finite number of discrete Fourier modes, $P(\vec{k}_i)$. We can organise the samples into shells of constant magnitude, $\bar{k}$ and width $\Delta k$, and count the number of modes (or samples) in each shell. In each shell we have $N(\bar{k}) \propto \bar{k}^2 \Delta k$ number of modes. We can see, then, that for large scales (small $k$) we have far fewer samples than on small scales (large $k$). We expect then to have large uncertainties on large scales.

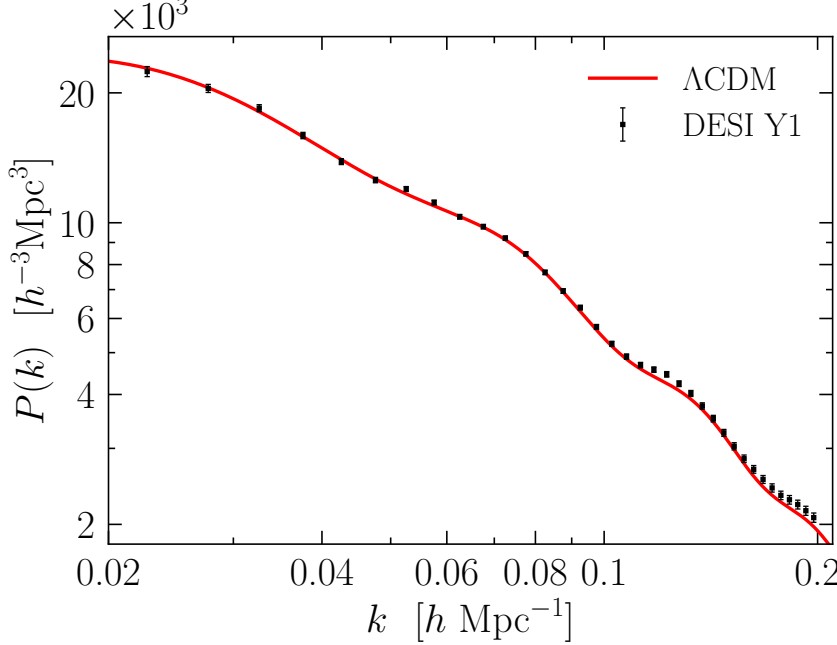

Figure 13: DESI Year 1 galaxy clustering measurements of the (reconstructed linear) matter power spectrum (black points) compared to the theory prediction of $\Lambda$CDM (red line) at $z = 0$ [46]

.

Another limitation to galaxy surveys is that we only have a discrete number of galaxies, unlike the density field that we model as a continuum. Indeed, in the observable Universe we expect about $10^{11}$ galaxies, but, in practice, we only observe a small fraction of them. When we build our galaxy density contrast, we are basing it on a number density which is, in practice

$$n_G(\vec{x}) = \sum_i \delta^3(\vec{x} - \vec{x}_i).$$

Given that this is the case, we have that the observed galaxy power spectrum, $P_G^{\text{obs}}(k)$, is related to the true power spectrum, $P_G(k)$, through

$$P_G^{\text{obs}}(k) = P_G(k) + P_{\text{shot}}(k),$$

where we can see there is a correction for what is known as 'shot noise' which, it can be shown, takes the form

$$P_{\text{shot}} = \frac{1}{\bar{n}_G}.$$

If $\bar{n}_G$ is too small, we can't detect $P_G(k)$. In general, we need to subtract out shot-noise to correct for the bias and, inevitably, it will lead to an addition source of uncertainty to cosmic variance.

Cosmic variance and shot noise are the main limitations of spectroscopic galaxy surveys for probing the matter density contrast. But there are others that need to be considered. For example, dust extinction will make it harder to detect fainter galaxies and so, in dusty regions of the sky, we may detect less galaxies and misinterpret what we see as an underdensity. Alternatively, we may confuse stars in the Milky Way for galaxies and thus artificially imprint Galactic structure onto the large scale structure of the Universe.

In Fig. 13 we show a comparison between the reconstructed linear power spectrum from the Dark Energy Spectroscopic Instrument (DESI) Survey [46] and the $\Lambda$CDM prediction.

## 15   The Cosmic Microwave Background

While spectroscopic surveys are, in some sense, the most direct way of accessing the large scale structure of the Universe, the CMB is, by far, the cleanest and is responsible for much of the success of cosmology.

Recall that Saha's equation tells you that, at $z \simeq 1100$, photons are released and can travel freely through space. They will have black body spectrum, with a temperature of $T_0 \simeq 0.13\,\text{eV} \simeq 2.735\,\text{K}$. Wien's law tells us that the peak frequency is related to the temperature, $\nu_{\text{peak}} \sim T$, and it will be redshifted as it propagates to us over the next 13 billion years. The photons will carry with them any residual inhomogeneity in the Universe, from the time they are released as well as along their flight path. The result is a slightly shifted black body spectrum in any direction, $\hat{n}$ in the sky and, if we measure these photons, we will end up with

$$T(\hat{n}) = T_0 \left[ 1 + \frac{\delta T}{T}(\vec{n}) \right].$$

We can use the expression we derived for the change in photon frequency along a perturbed geodesic to find

$$\frac{\delta T}{T}(\vec{n}) \;\; = \;\; \frac{1}{4}\delta_\gamma(\Delta\tau_*\hat{n},\tau_*) + \Phi(\Delta\tau_*\hat{n},\tau_*) - \vec{n}\cdot\vec{v}(\Delta\tau_*\hat{n},\tau_*) + 2\int_{\tau_*}^{\tau_0} d\tau\,\Phi'(\Delta\tau\hat{n},\tau),$$

where $\tau_*$ is the comoving time at recombination, $\Delta\tau_* = \tau_0 - \tau_*$ ($\Delta\tau = \tau_0 - \tau$), and we have used the fact that the photons obey the Stefan-Boltzmann law, $\rho_\gamma \propto T^4$.

We can see what it looks like in Fig.1. The angular power spectrum of the CMB is dominated by a series of peaks and troughs [47]. These oscillations are a remnant of pre-recombination physics in which the photons were tightly coupled to the baryons and behaved, collectively, as a fluid. The scale of the oscillations are set by the sound horizon size at the end of recombination, when the photons last scattered off the baryons, leading to the comoving sound horizon, $r_*$. While these are exactly the same oscillations imprinted in the baryons which then manifest themselves as the BAOS, there the scale is slightly different, and is denoted (as we saw) by $r_d$; the differences are of order 5% and are due to the fact that the Baryons decouple from the Compton drag of the photons slightly later (at $z = 1050$ as opposed to z=1100).

We can define the angular scale of these peaks as $\theta_* = r_*/\chi_*$, where $\chi_* = \Delta\tau_*$. From current measurements of the CMB, we have that $\theta_* \simeq 0.6^{\text{o}}$. The peak positions are harmonics of this scales are given by $\ell^{\text{peaks}} \simeq n\pi/\theta_*$. Given that these quantities depend on $\chi_*$, they are a direct probe of the angular diameter distance to the CMB depend on cosmological parameters, as we saw previously [48, 49].

If we now look at the large angular scales (small $\ell$), we are looking at scales which are directly related to the primordial power spectrum. The scalar spectral index, $n_s$, tells us the overall slope of that line – indeed, we have that $\ell^2 C_\ell \propto \ell^{n_s-1}$. There is an additional correction due to the last part in Eq. (4), known as the Integrated Sachs-Wolfe (ISW) effect [50]. We have, in a purely matter dominated Universe, that $-k^2\Phi \sim \bar\rho a^2 \delta \sim a^{-3} \times a^2 \times a \sim$ constant. This means that $\Phi' = 0$ and there is no ISW. If there is a period of $\Lambda$ domination, then $\delta \neq a$ and $\Phi$ will be time varying. Thus the ISW is a direct probe of the presence of $\Lambda$, independent from the background evolution [51].

Finally, if we look at small angular scales (i.e. large $\ell$'s), we see that the angular power spectrum is damped. This is a result of the damping of perturbations in the baryon photon fluid during the transition from tight-coupling to free-streaming during recombination, the process we described above as Silk damping [16].

All of this structure in the CMB gives us much information about the cosmological model, in particular, about the cosmological parameters. This has been done to great effect with the recent generation of CMB satellite missions – WMAP [52] and Planck [53]– and with the new generation of high resolution, ground based experiments, SPT [54] and ACT [55].

The CMB is also polarised due to the anisotropic nature of Thomson scattering [56]. During recombination and, later, during reionisation temperature fluctuations will source polarisation. The polarisation of the CMB can be described as a spin-2 field, or alternatively as a vector field with peculiar transformation properties. As such, we can decompose it in the same way we decompose any other vector field, into a irrotational, or curl-free, part – an 'E-mode' – and a rotational, or divergenceless, part – a 'B-mode' [57, 58]. The B-mode has the particularity of not being sensitive to scalar perturbations, i.e., the linear density fluctuations which we have been focussing on in these lectures. It will be sensitive to non-linear processes that may transform the linear density perturbations (such as the weak lensing of the CMB, which we will refer to in the next section) and primordial gravitational waves. There is an active observational campaign that is trying to detect the B-modes but it is far more challenging than measuring the temperature fluctuations; while we have that the angular power spectrum of temperature fluctuations is of order $\ell(\ell+1)C^{TT}\ell \sim 10^3(\mu\mathrm{K})^2$, we have that $\ell(\ell+1)C^{EE}\ell \sim 10^{-1}(\mu\mathrm{K})^2$ and, most challenging of all, $\ell(\ell+1)C^{BB}\ell \sim 10^{-5}(\mu\mathrm{K})^2$.

There are a number of uncertainties in characterising the CMB. The first one we have already referred to: at large scales (small $\ell$) constraints are hampered by cosmic variance. On small scales, instrumental noise plays an important role. If we think of our measurement as a pixelised sky map, each pixel has a finite number of measurements. The smaller the scales we want to probe, the smaller the size of pixel we need to use and the fewer measurements – each pixel will be noisier. Furthermore, the telescope we use will have a finite resolution, typically given by a beam-size full-width half maxiumum, $\theta_{\mathrm{FWHM}}$. The end result is we have an uncertainty in the angular power spectrum of the CMB of the form [59]

$$\Delta C_\ell = \sqrt{\frac{2}{(2\ell+1)f_{\mathrm{sky}}}}\left(C_\ell + N_\ell\right),$$

where $f_{\mathrm{sky}}$ is the sky fraction covered, and $N_\ell$ is the effective noise power spectrum of observation which can be decomposed as

$$N_\ell = \Delta_N^2 \exp\left(\ell(\ell+1)\sigma_B^2\right), \tag{4}$$

where $\Delta_N$ is the noise sensitivity per pixel and $\sigma_B = \theta_{\mathrm{FWHM}}/\sqrt{8\ln 2}$.

There is an added complication which is of a more insidious and systematic nature and which is becoming the dominant source of uncertainty as we pursue ever more faint signals (such as the B-modes): foregrounds [60]. By this we mean sources of microwave photons

other than the surface of last scattering. One can divide these contaminants between local – specifically the Galaxy – and cosmological or extra-Galactic. The Galactic contaminants are; *Synchrotron emission* – cosmic rays spiralling in the Galactic magnetic field, dominant at low frequencies (below 100 GHz); *Free-Free emission* – electrons scattering off hot ionised gas, also at low frequencies; *Dust emission* – interstellar dust grains heated by starlight, dominant at high frequencies (above 100 GHz); *Anomalous microwave emission* – spinning dust grains emitting dipole radiation, peaking at 20-40 GHz. The extra-Galactic contaminants are: *Radio and Infrared Point Sources* – Active galactic nuclei (AGN) and star-forming galaxies, dominant at low frequencies (below 100 GHz); *Cosmic Infrared Background (CIB)* – Unresolved emission from distant dusty star-forming galaxies, dominant at high frequencies (above 200 GHz); *Sunyaev-Zeld'ovich (SZ) Effects* – Inverse Compton scattering of CMB photons off hot electrons in galaxy clusters (thermal SZ) and Doppler shift from bulk motion of ionised gas (kinetic SZ). Almost all of these foregrounds (except for the kSZ) do not have black body spectra. Thus multifrequency measurements can be used to disentangle between them and the CMB. On the other hand, the extragalactic foregrounds should trace the large scale structure of the Universe and can, in principle, be used to gather more information about it than the CMB on its own.

# 16   Weak lensing

We saw, from the geodesic equation, that trajectories of photons will be modified by intervening fluctuations in the metric. This is what is known as *gravitational lensing*. In a cosmological setting, where the gravitational potential, $\Phi$, fluctuations are small, it is known as *weak lensing* [1, 61].

Consider a localised mass, like a galaxy, a cluster or any overdensity of matter. It will generate fluctuations in the $\Phi$. A photon emitted from a distant object (such as a galaxy) will be deflected as it passes through this fluctuation and its angular position in the sky, $\vec{\theta}_o$ will be shifted relative to its true angular position, $\vec{\theta}$, $\delta\vec{\theta} = \vec{\theta}_o - \vec{\theta}$. We have that

$$\delta\vec{\theta} = 2\int_0^{\chi_S} d\chi \left(1 - \frac{\chi}{\chi_S}\right) \vec{\nabla}_\perp \Phi(\chi).$$

It is useful to define the lensing potential, $\Phi_L$

$$\Phi_L = 2\int_0^{\chi_S} \frac{\chi_S - \chi}{\chi \chi_S} \Phi(\chi).$$

We can then re-express the deflection angle in terms of the transverse derivative on the sky

$$\delta\vec{\theta} = \vec{\nabla}_\theta \Phi_L.$$

Consider now an extended object and measure the angular position of two points, A and B, on that object. The observed separation between the two points is given by

$$\Delta\vec{\theta}_o = \Delta\vec{\theta} + \vec{\nabla}_\theta \Phi_L|_A - \vec{\nabla}_\theta \Phi_L|_B \simeq \Delta\vec{\theta} + \mathcal{H}_L \Delta\vec{\theta},$$

where we have Taylor expanded the last two terms and have defined the resulting Hessian matrix

$$\mathcal{H}_L \equiv \frac{\partial^2 \Phi_L}{\partial \theta_i \partial \theta_j} \equiv \begin{pmatrix} \kappa + \gamma_1 & \gamma_2 \\ \gamma_2 & \kappa - \gamma_1 \end{pmatrix},$$

where $\kappa$ is the convergence and $\gamma_i$ is the shear.

The convergence and shear will affect an extended object in different ways. To see this, consider an object with an intensity profile $I(\vec{\theta})$. We can work out the observed flux, $F_o$ in terms of the true flux, $F$

$$F_o = \int d^2\theta_o I(\theta_o) = \int d^2\theta^2 \det(1 + \mathcal{H}_L)I(\theta) = (1 + 2\kappa)F.$$

Thus we see that the convergence changes the brightness of the source.

Now consider the quadropole moment of the object (which captures information about its shape)

$$Q_o^{ij} = \int d^2\theta_o \Delta\theta_o^i \Delta\theta_o^j I(\theta_o).$$

We can define the ellipiticities

$$\vec{e}_o = \frac{1}{\text{tr}[Q_o]}\left(Q_o^{11} - Q_o^{22}, Q_o^{12}\right),$$

which are easily related to the shear; if the object is (originally) circular, we have that $e_o^i = 2\gamma_i$.

We can use measurements of the ellipticities to place constraints on the shear by measuring the shapes of galaxies. Galaxies are *not* circularly symmetric on the sky – a simple approximation is to consider that they are all intrinsically elliptical, to some extent. However their intrinsic ellipticities are fundamentally completely uncorrelated between galaxies and if we average over a large enough sample, we can get rid of these intrinsic ellipticities. We then have the ellipticities induced by the gravitational lensing will be correlated with each other as they are generated by the large scale structure of the Universe which has long range correlations (described by the power spectrum of matter perturbations). Therefore, if we measure the angular power spectrum of the shear (via the ellipticities) we will constrain the matter power spectrum [62].

The convergence and the shear will depend on the underlying density field, that is causing the weak lensing, in a qualitatively similar manner. For simplicity, if we focus on the convergence, we have that, for an individual source, it can be expressed in a familiar form

$$\kappa_S(\hat{n}) = \int_0^{\chi_S} d\chi \, q_{SL}(\chi)\delta_M(\chi\hat{n}, \chi),$$

where the kernel is

$$q_{SL} = \frac{3}{2}\Omega_M H_0^2 \frac{\chi}{a(\chi)}\frac{\chi_S - \chi}{\chi_S}.$$

If we have a distribution of sources, with a redshift distribution, $p(z)$ (which can be expressed as a function of $\chi_S$), the convergence will be

$$\kappa(\hat{n}) = \int d\chi_S p(\chi_S)\kappa_{SL}(\hat{n}).$$

Alternatively, we have that

$$\kappa(\hat{n}) = \int d\chi \, q_L(\chi)\delta_M(\chi\hat{n}, \chi),$$

where

$$q_L(\chi) = \int d\chi_S p(\chi_S)q_{SL}(\chi).$$

The lensing kernel, unlike the kernel for spectroscopic galaxy surveys, is very broad and will project a broad range of three dimensional modes onto any particular angular scale. We can, once again, use the Limber approximation, presented above to connect the matter power spectrum with the lensing power spectrum.

There are a number of complications in extracting weak lensing information. For a start, it is very difficult to measure ellipticities of galaxies. Often, the telescope resolution is quite poor and any given galaxy will be covered by, at most, a handful of pixels. To mitigate this measurement error, weak lensing surveys try to target a very large number of source galaxies, around $10^9$ as opposed to $10^{6-7}$ for spectroscopic surveys. This, in turn, means that it is impossible to pin down the exact redshift of each source galaxies and quicker, dirtier, methods are used that use photometric methods to get a rough idea of the redshifts (known as *photometric redshifts*) and, in turn, their comoving distances. These methods are imperfect and can lead to biases when trying to reconstruct the matter power spectrum from measurements of the angular power spectrum of weak lensing. The flip side is that these measurements are not plagued by galaxy bias and are a direct probe of the matter density distribution. Finally, a key assumption is that the intrinsic ellipticities of galaxies are completely uncorrelated and random. Unfortunately this is not necessarily the case. Galaxies form in the ambient large scale distribution of matter and their properties and, more crucially, orientations, may be influenced by, for example, the ambient tidal field. These effects lead to what is known as intrinsic alignments [63,64],which may be confused with the weak lensing correlations which one is trying to determine. Conversely, it may be possible to use these intrinsic alignment correlations themselves to place constraints on the large scale structure of the Universe [65].

# 17 Statistical inference in cosmology

We now have a mathematical model and we have data. To put them together, we need to use some form of statistical inference. The method of choice is Bayesian inference is based on the judicious use of Bayes theorem [66,67]. In the specific case of cosmology we have data, $\mathcal{D}$, and a model described in terms of a set of parameters, $\vec{\alpha}$. Bayes' theorem tells us that the values of the parameters given the data are

$$P(\vec{\alpha}|\mathcal{D}) = \frac{P(\mathcal{D}|\vec{\alpha})P(\vec{\alpha})}{P(\mathcal{D})},$$

where $P(\vec{\alpha}|\mathcal{D})$ is the posterior (what we want to determine), $P(\mathcal{D}|\vec{\alpha})$ is the *likelihood* of the data, given the specific choice of model parameters, $P(\vec{\alpha})$ is the prior which encapsulates our prior beliefs on the parameters we are trying to determine and $P(\mathcal{D})$ is the evidence of the model, given the data.

We use our knowledge about the observables to construct the likelihood – in the case of the background cosmology we need to fold in the observational uncertainties in to construct, for example a goodness of fit, or $\chi^2(\vec{\alpha}, \mathcal{D})$ for the data given the model, and we can approximate the likelihood by assuming that

$$P(\mathcal{D}|\vec{\alpha}) \propto \exp\left[-\frac{1}{2}\chi^2(\vec{\alpha}, \mathcal{D})\right].$$

More detailed knowledge about the observables and uncertainties allow us to build more accurate and elaborate likelihoods [1]. We then need to make certain assumptions about the values of the parameters, either making as few assumptions about their values and ranges or folding in prior knowledge which may come from other experiments.

The end result will be a function which is multivariate (in $\vec{\alpha}$) which we can then explore. We can look at its maximum but, more importantly, we can take slices such that 68%, 95%,

or more of the volume lie within isocontours. These contours will delimit our confidence regions for these parameters, they will tell us what ranges of parameters are allowed by a given collection of data, i.e. which mathematical model, with what choice of parameters, is the most probable explanation of the data one is considering.

A range of tools have been developed for undertaking this statistical inference. For a start, one needs efficient methods for calculating the predictions of the mathematical model that can then be compared to the data. Notable examples are fast Boltzman solvers such as `CAMB` [68] and `CLASS` [69] for the linear regime, and approximation methods for the non-linear regime, involving perturbation theory [70], the halo model [71] or approximation functions (or *emulators*) [72,73]. But also, one needs to use efficient methods for exploring the posterior function, to not only find its maximum but to also, accurately map out its shape. Typically one uses stochastic methods, from straightforward Monte Carlo Markov Chains (MCMC) [74] and nested samplers [75] to directed, gradient based methods that use differential information about the posterior, such as for example Hamiltonian Markov Chains [76].

Fast and accurate methods for exploring the posterior are becoming increasingly important as the models one considers become ever more complex. Not only must the include the (cosmological) parameters that one is trying to constrain but also parameters that capture the role of astrophysical processes that might, for example, contaminate the non-linear evolution of structure, or the foregrounds that might affect CMB measurements, or the uncertainties in the distributions of galaxies in weak lensing surveys [77]. The more parameters, the harder it is to accurately characterise the posterior distributions of the cosmological parameters.

## 18 Constraining the cosmological model

We have now sketched all the ingredients that go into building a mathematical model of the Universe, finding its observables, observing the Universe and inferring its properties. Over the last three decades we have progressed at an astounding pace, finding constraints on the cosmological parameters that we are unimaginable when we set of on this endeavour. For example, by the end of the 1990s, we had no idea what the geometry of the Universe, $\Omega_k$ was. We now know it to within $10^{-3}$.

We seem to have converged on a very well-characterised model, $\Lambda$CDM, which seems to be, mostly, robust to different measurements. There are wrinkles – inconsistencies or *tensions* as they are now called – but it is premature to believe that they are causing trouble. Thus, in this last section we will go through current constraints on the parameters of this mathematical model, what we call *cosmological parameters*.

*Hubble constant $H_0$*: It has become the norm to distinguish between 'early universe' constraints from the CMB and 'late universe' constraints from the distance ladder. The CMB consistently gives a lower value, with the Planck 2018 measurements [53] giving $H_0 = 67.4 \pm 0.5$ km s$^{-1}$ Mpc$^{-1}$. This is a $\sim 5\sigma$ tension with locally inferred value of the SH0ES collaboration [36], which uses (primarily) Cepheids and and SN I$_a$ to find $H_0 = 73.2 \pm 1.0$ km s$^{-1}$ Mpc$^{-1}$. This dichotomy between early and late universe constraints is muddied by measurements from the Chicago-Carnegie Hubble Programme (CCHP) [38] using the TRGB method which find $H_0 = 68.81 \pm 3.0$ km s$^{-1}$ Mpc$^{-1}$. While it does have larger uncertainties than other local measurements, it does tend towards the early universe constraints. This tension between early and late time constraints is a source of vigorous debate.

*Curvature of the Universe $\Omega_k$*: Over the past decades, constraints have converged, remarkably onto a Euclidean universe. A combination of constraints from Planck and the DESI spectroscopic redshift survey [78] give us $\Omega_k = -0.0007 \pm 0.0019$, i.e. spatial flatness to within 0.1%.

*Scalar Spectral Index $n_S$:* Constraints from a Planck 2018 , give $n_S = 0.965 \pm 0.004$ which is almost 9-$\sigma$ away from scale invariance $n_S = 1$. This is of particular interest because inflation, as a theory of initial conditions, requires slight deviations from scale invariance to be self consistent. More recent constraints on $n_S$ from the ACT experiment [55] have nudged the mean value slightly higher to $n_S = 0.974 \pm 0.003$, but still distinct from scale invariance.

*Total Matter Density $\Omega_M$:* There is broad agreement on this parameter with the Planck 2018 constraints, leading to $\Omega_M = 0.315 \pm 0.007$. Nevertheless, some mild tensions have been emerging, depending on the analysis. For example, assuming $\Lambda$CDM, the DESI spectroscopic survey, on its own leads to $\Omega_M = 0.298 \pm 0.009$, which is marginally inconsistent with the CMB constraints. A compilation of weak lensing measurements [79], leads to a further discrepancy, with $\Omega_M = 0.212^{+0.017}_{0.032}$.

*Fractional Baryon Density $\Omega_b h^2$:* Current CMB constraints from Planck 2018 give $\Omega_b h^2 = 0.02237 \pm 0.00015$ which are very consistent with constraints from big bang nucleosynthesis [80]: $\Omega_b h^2 = 0.0219 \pm 0.0003$. There are still slight discrepancies with constraints from the abundance of Lithium, but these are within current uncertainties.

*Amplitude of density fluctuations $\sigma_8$:* The CMB places tight unambiguous constraints, with the Planck 2018 measurements, of $\sigma_8 = 0.811 \pm 0.006$, while spectroscopic redshift surveys, with DESI, lead to $\sigma_8 = 0.8121 \pm 0.005$. Weak lensing surveys, on the other hand lead to slightly lower values; it has become customary to use $S_8 = \sigma_8 (\Omega_M/0.3)^{1/2}$. The Planck 2018 constraints give $S_8 = 0.832 \pm 0.013$. The DES year 3 analysis [81] found $S_8 = 0.776 \pm 0.017$, while the KIDS analysis [82] has $S_8 = 0.776 \pm 0.03$, although a more recent, more complete KIDS-Legacy analysis [83] has found a slightly higher value, $S_8 = 0.821^{+0.016}_{-0.021}$.

*Fractional Dark Energy Density $\Omega_{DE}$:* Current constraints for $\Lambda$CDM from the DESI spectroscopic survey combined with the CMB are $\Omega_\Lambda = 0.687 \pm 0.004$. If one relaxes the assumption about the type of dark energy and uses, for example, the CPL parametrisation for the equation of state, on has $\Omega_{DE} = 0.647 \pm 0.021$. If one, in addition, uses the DES Y5 SN $I_a$ data, one finds $\Omega_{DE} = 0.681 \pm 0.006$.

*Dark Energy Equation of State w:* For a number of years, the $\Lambda$CDM model has been consistent with observational data. It is still, at many levels, the simplest and preferred model. If one relaxes the assumption that $w = -1$ and allows it to be a free, constant paramenter, a combination of CMB, BAO and SN $I_a$ data pin it down to the $\Lambda$CDM value. If one allows for a time evolving equation of state, using the CPL parametrisation, one has that DESI alone find $w_0 = -0.48^{+0.35}_{-0.17}$ and $w_a < -1.34$, i.e. evidence for evolving, dark energy. If one then combines it with CMB and SNI $I_a$ data from the DES Year 5 analysis [84], the evidence strengthens to $w_0 = -0.752 \pm 0.057$ and $w_a = -0.86^{+0.23}_{-0.20}$. This is a striking result and should be handled with care. It is highly dependent on the low redshift data in the DES Y5 survey and we should wait and see, while the quality of the data develops.

*Neutrino Masses $\sum m_\nu$:* Current constraints on neutrino masses are highly dependent on the CMB pipeline used to extract parameter constraints. Current upper bounds on the sum of neutrino masses range between 0.064 to 0.07 [55]. These bounds are low and marginally consistent with minimal neutrino mass scenarios (normal hierarchy). If one includes a time dependent equation of state, one has considerably looser bounds, $\sum m_\nu < 0.13$.

It is clear that we have entered an era of what has been dubbed *precision* cosmology. The uncertainty on the cosmological parameters has plummeted and we are reaching a point of diminishing returns. Nevertheless we have also entered a phase in which what seem to be minor inconsistencies, or tensions, are driving the field forward. $\Lambda$CDM has stood the test of time (so far!) but are we at the point in which we need to dig deeper and reassess some its core assumptions? Fortunately, at time of writing, we have just entered a new glorious phase of observation cosmology, the 'Stage IV' era in which a host of new powerful observatories are

coming online. We look forward to what the data will tell us over the next decade: is ΛCDM the ultimate theory or is there another, yet more mysterious explanation?

## Acknowledgements

We thank the organisers of the 2025 School on the Dark Universe, Phillipe, Sandrine, Guilhem and Pauline for inviting us for such an enjoyable event. We acknowledge the relentless wisdom and patience of David Alonso for helping us understand some of the intricacies of modern survey science and for sharing his embryonic cosmology textbook with us. We thank Josu Aurrekotxea for producing Fig. 1 and the DESI collaboration and Rasa Cereskaite for providing the data used in Fig.13.

**Funding information** PGF acknowledges support from the Beecroft Trust and STFC, AR acknowledges the support of an STFC Studentship.

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
