# Peer review of "A Short Introduction to Cosmology and its Current Status"

_SciPost Physics Lecture Notes_

## Round 1 · Referee Report · María Pérez Garrote (Referee 1) · 2025-10-29

Report

These lecture notes provide a detailed summary of all the material covered in the talks of the Les Houches Summer School 2025. As stated in the preamble, they offer a good overview of the current status of the modern cosmology and many of the relevant topics in this field are discussed in sufficient detail to provide a comprehensive picture of this area.

Requested changes

1- The equations are not numbered, it is clearer for the reader if they are numbered in case any of them need to be referenced. 2- On page 4, the energy momentum tensor corresponds to a perfect fluid. It is important to specify that it does not describe a general fluid. 3- On page 9, note that the relation between the luminosity distance and the angular distance is commonly referred to as the distance duality relation, and it holds if the spacetime is described by a pseudo-Riemannian manifold, photons propagate along null geodesics and the number of photons is conserved. 4- On page 12, in the expressions for the quintessence field’s energy density and pressure, it should be $\psi$ instead of $\phi$. 5- On page 19, indicate that for the scalar perturbed metric the Newtonian gauge has been chosen. 6- On page 20, in the sentence "we have that the Newton Poisson equation is a good approximation when ...", it should be $k\tau \gg 1$ instead of $k\eta \gg 1$. 7- On page 35, it should be $P^\mu\propto \frac{dx^\mu}{d\lambda}$. 8- On page 41, in the definition of the lensing potential $\Phi_L$ the differential $d\chi$ is missing.

Recommendation

Ask for minor revision

---

## Round 1 · Referee Report · Anonymous (Referee 2) · 2025-11-1

Report

These lecture notes have successfully achieved the difficult challenge of giving an exhaustive overview of modern cosmology in a limited amount of time and space. They manage to cover a lot of different topics, while remaining simultaneously accessible enough to the novice reader, yet diving into quantitative details. I especially appreciate the authors' choice of sticking to the basics in each section, yet concluding each section with a list of all the caveats that the simple model can suffer from, allowing the reader to be aware of all the hidden complexity that can exist without losing them into too much detail.
As a reviewer, I was asked to provide a demanding report, as these notes are intended to be part of the historical Les Houches lectures, which aim to remain accessible in many years from now. That is why, although the notes are of great quality, I would like to suggest quite many changes to the text that I detail in the next part of the present report.

Requested changes

As these notes are intended for the non-expert reader, I have especially focused on suggesting pedagogical improvements for the notes wherever it felt necessary.

1) The transitions when switching between topics (both within a given section or between consecutive sections) are often quite abrupt. More logical links between statements should be included throughout the text to improve the logical flow of the paper, see e.g. before the beginning of section 14.

2) These notes are written in UK English. Although I have no problem with this, I would simply like to point out that these notes will belong to a compilation of lectures of which some may be written in US English. Maybe the editors have some requirements about this that the authors should check.

3) Throughout the document, the notation for the conformal time switches back and forth between eta and tau. A single consistent convention should be chosen.

4) Equations have not been given numbers. First, I think it is preferable for the reader that each of them is given a number so they can be referred to more easily. Second, it made it difficult for the report to refer to specific locations in the text. That is why, instead of listing in this textbox all the specific changes I want to suggest, I have attached a pdf document to the report which goes over the notes with added comments in the margins. The comments in the pdf are color-coded as follows:

5) Red comments are critical remarks and correspond to passages of the notes which require modifications. Orange comments are strongly advised suggestions to the authors, but which they can decline if they think the option they have chosen was more relevant. Blue comments are typographical/grammatical errors. On this last point, I encourage the authors to read through the entire document again as I was likely not able to spot all these errors.

Attachment

Recommendation

Ask for minor revision

---

## Editorial Decision

resubmitted